# $\alpha_1$-Adrenergic receptor–PKC–Pyk2–Src signaling boosts L-type Ca$^{2+}$ channel Ca$_V$1.2 activity and long-term potentiation in rodents

Kwun Nok Mimi Man[1], Peter Bartels[1†], Peter B Henderson[1†], Karam Kim[1], Mei Shi[2], Mingxu Zhang[1,2], Sheng-Yang Ho[1], Madeline Nieves-Cintron[1], Manuel F Navedo[1], Mary C Horne[1,2*], Johannes W Hell[1,2*]

[1]Department of Pharmacology, University of California, Davis, United States; [2]Department of Pharmacology, University of Iowa, Iowa City, United States

**Abstract** The cellular mechanisms mediating norepinephrine (NE) functions in brain to result in behaviors are unknown. We identified the L-type Ca$^{2+}$ channel (LTCC) Ca$_V$1.2 as a principal target for G$_q$-coupled $\alpha_1$-adrenergic receptors (ARs). $\alpha_1$AR signaling increased LTCC activity in hippocampal neurons. This regulation required protein kinase C (PKC)-mediated activation of the tyrosine kinases Pyk2 and, downstream, Src. Pyk2 and Src were associated with Ca$_V$1.2. In model neuroendocrine PC12 cells, stimulation of PKC induced tyrosine phosphorylation of Ca$_V$1.2, a modification abrogated by inhibition of Pyk2 and Src. Upregulation of LTCC activity by $\alpha_1$AR and formation of a signaling complex with PKC, Pyk2, and Src suggests that Ca$_V$1.2 is a central conduit for signaling by NE. Indeed, a form of hippocampal long-term potentiation (LTP) in young mice requires both the LTCC and $\alpha_1$AR stimulation. Inhibition of Pyk2 and Src blocked this LTP, indicating that enhancement of Ca$_V$1.2 activity via $\alpha_1$AR–Pyk2–Src signaling regulates synaptic strength.

*For correspondence:
mhorne@ucdavis.edu (MCH);
jwhell@ucdavis.edu (JWH)

†These authors contributed equally to this work

## Editor's evaluation

This study reports of a new signaling pathway in hippocampal neurons by which $\alpha_1$ receptors for norepinephrine regulate Cav1.2 calcium channels; activation of $\alpha_1$ receptors enhances a form of long-lasting synaptic plasticity that is dependent on L-type calcium channels. The experiments are comprehensive and well-executed, and the main conclusions are compellingly supported by the data shown. The work has significance for the field of neuroscience in general and for cellular mechanisms of neuroregulation in particular.

## Introduction

Norepinephrine (NE) causes arousal and augments behavioral acuity and learning (*Berman and Dudai, 2001*; *Cahill et al., 1994*; *Carter et al., 2010*; *Hu et al., 2007*; *Minzenberg et al., 2008*). NE signals via the G$_q$-coupled $\alpha_1$-adrenergic receptor (AR), G$_i$-coupled $\alpha_2$AR, and G$_s$-coupled $\beta_1$, $\beta_2$, and $\beta_3$ ARs. $\beta$ARs act through adenylyl cyclase (AC), cAMP, and PKA (*Sanderson and Dell'Acqua, 2011*). The $\beta_2$AR, G$_s$, AC, and PKA are all associated with the L-type Ca$^{2+}$ channel (LTCC) Ca$_V$1.2 for efficient signaling in neurons (*Davare et al., 2001*; *Dittmer et al., 2014*; *Murphy et al., 2014*; *Oliveria et al., 2007*; *Patriarchi et al., 2016*; *Qian et al., 2017*) and heart (*Balijepalli et al., 2006*). The formation of this signaling complex identifies Ca$_V$1.2 as a major effector of signaling by NE. We now find that Ca$_V$1.2 is also a major effector for signaling via the $\alpha_1$AR, which has a higher affinity for NE than

βARs (*Giustino and Maren, 2018*; *Ramos and Arnsten, 2007*). Importantly, a large body of evidence implicates the $\alpha_1$AR in NE's role in attention and vigilance (*Bari and Robbins, 2013*; *Berridge et al., 2012*; *Hahn and Stolerman, 2005*; *Hvoslef-Eide et al., 2015*; *Liu et al., 2009*; *Puumala et al., 1997*; *Robbins, 2002*).

Ca$_V$1.2 fulfills a remarkably broad spectrum of functions. Dysfunctions due to mutations in Ca$_V$1.2 span from impaired cardiac contractility to the autistic-like behaviors seen in Timothy syndrome (*Splawski et al., 2004*). Furthermore, Ca$_V$1.2 has been linked to filopodia formation in invasive cancer cells (*Jacquemet et al., 2016*). Ca$_V$1.2 is by far the most abundant LTCC in heart and accounts for ~80% of all LTCCs in brain (*Hell et al., 1993a*; *Sinnegger-Brauns et al., 2004*). It governs the heartbeat, vascular tone, and neuronal functions including long-term potentiation (LTP) (*Ghosh et al., 2017*; *Grover and Teyler, 1990*; *Moosmang et al., 2005*; *Patriarchi et al., 2016*; *Qian et al., 2017*), long-term depression (*Bolshakov and Siegelbaum, 1994*), neuronal excitability (*Berkefeld et al., 2006*; *Marrion and Tavalin, 1998*), and gene expression (*Dolmetsch et al., 2001*; *Li et al., 2016*; *Li et al., 2012*; *Ma et al., 2014*; *Marshall et al., 2011*; *Murphy et al., 2014*; *Wheeler et al., 2012*). Studies on Ca$_V$1.2 mutant mice suggest that this channel plays a central role in anxiety disorders, depression, and self-injurious behavior (*Sinnegger-Brauns et al., 2004*). Congruently, LTCC blockers elicit antidepressant effects while agonists induce depression-like behavior (*Mogilnicka et al., 1987*; *Mogilnicka et al., 1988*) and self-biting in mice, a symptom associated with autism (*Jinnah et al., 1999*).

Ca$_V$1.2 consists of the pore-forming subunit $\alpha_1$1.2, a β subunit and the $\alpha_2\delta$ subunit (*Catterall, 2000*; *Dai et al., 2009*; *Zamponi et al., 2015*). The β and $\alpha_2\delta$ subunits facilitate release of $\alpha_1$1.2 subunits from the endoplasmic reticulum, inhibit ubiquitin-mediated degradation of voltage-gated calcium channels, influence electrophysiological properties of Ca$^{2+}$ channels, such as activation and inactivation, and play diverse roles in the regulation of these channels (*Catterall, 2000*; *Dai et al., 2009*; *Zamponi et al., 2015*).

In the cardiovascular system, the $\alpha_1$AR, the endothelin receptor ET1, and the angiotensin receptor AT$_1$ are important regulators of LTCC currents via G$_q$ signaling (*Catterall, 2000*; *Kamp and Hell, 2000*; *Voelker et al., 2023*). G$_q$ stimulates phospholipase C-β to induce production of diacylglycerol (DAG) and inositol-1,4,5-trisphosphate (IP$_3$), which triggers Ca$^{2+}$ release from intracellular stores. DAG and Ca$^{2+}$ act in concert with phosphatidyl-serine to activate different PKC isoforms. Stimulation of PKC mostly leads to an increase in Ca$_V$1.2 activity (*Bkaily et al., 1995*; *Dai et al., 2009*; *Döşemeci et al., 1988*; *He et al., 2000*; *Kamp and Hell, 2000*; *Lacerda et al., 1988*; *Navedo et al., 2005*). However, an inhibitory effect of PKC on Ca$_V$1.2 currents has been reported in cardiomyocytes (*Cheng et al., 1995*; *Voelker et al., 2023*). This inhibition is mediated by phosphorylation of residues T27 and T31 by PKC in an isoform of $\alpha_1$1.2 that is expressed in heart (*McHugh et al., 2000*). T27/T31 are not present in the most prevalent brain isoform due to alternative splicing (*Snutch et al., 1991*); thus, the inhibitory effect of PKC on LTCC currents is typically absent in neurons and neural crest-derived PC12 cells, or in vascular smooth muscle (*Navedo et al., 2005*; *Taylor et al., 2000*). Here, we show that stimulation of the $\alpha_1$AR and of PKC consistently augments LTCC in hippocampal neurons.

Despite the prominent role of PKC in augmentation of Ca$_V$1.2 activity, how PKC mediates this effect has been unknown. PKC activates the nonreceptor tyrosine kinase Pyk2, a signaling process first shown in PC12 cells (*Dikic et al., 1996*; *Lev et al., 1995*) and later primary neurons (*Bartos et al., 2010*; *Huang et al., 2001*), and cardiomyocytes (*Sabri et al., 1998*). Activation of PKC triggers autophosphorylation of residue Y402 on Pyk2 to create a binding site for the SH2 domain of Src, which upon binding to Pyk2 becomes activated (*Dikic et al., 1996*). Src increases LTCC activity in smooth muscle cells (*Gui et al., 2006*; *Hu et al., 1998*; *Wu et al., 2001*), retinal pigment epithelium (*Strauss et al., 1997*), and neurons (*Bence-Hanulec et al., 2000*; *Endoh, 2005*; *Gui et al., 2006*). Furthermore, PKC (*Navedo et al., 2008*; *Yang et al., 2005*) and Src (*Bence-Hanulec et al., 2000*; *Chao et al., 2011*; *Hu et al., 1998*) are physically and functionally associated with Ca$_V$1.2. These findings underscore the physiological relevance of Src in regulating Ca$_V$1.2. Importantly, the pathway by which Src is activated in the context of Ca$_V$1.2 regulation has not been determined.

Once we established that stimulation of PKC or the G$_q$/PKC-coupled $\alpha_1$AR strongly augments LTCC activity in neurons, we tested whether Pyk2 mediates this upregulation of channel activity. We link the $\alpha_1$AR–PKC signaling to Src, which thus emerges as an important mediator of tyrosine phosphorylation on Ca$_V$1.2 downstream of G$_q$-coupled receptors. In neurons, the nearly twofold increase in LTCC currents upon stimulation of PKC with phorbol-12-myristate-13-acetate (PMA) or via the $\alpha_1$AR was

blocked by inhibitors of Pyk2 and Src, consistent with earlier data showing that Src elevates $Ca_V1.2$ activity to a comparable degree (*Bence-Hanulec et al., 2000*; *Gui et al., 2006*). Furthermore, we found that Pyk2 co-immunoprecipitated with $Ca_V1.2$ in parallel to Src. We identified the loop between domains two and three of $\alpha_11.2$ as the Pyk2-binding site. Stimulation of PKC either directly with PMA or through the $G_q$-coupled bradykinin (BK) receptor leads to tyrosine phosphorylation of $\alpha_11.2$ in PC12 cells. Abrogation of Pyk2 or Src activity ablated the phosphorylation. Finally, we discovered that the LTP in young mice mediated by LTCC-dependent $Ca^{2+}$ influx during 200 Hz tetani (termed $LTP_{LTCC}$) that is not NMDAR dependent, required $\alpha_1AR$ stimulation and both Pyk2 and Src activity. These findings implicate upregulation of $Ca_V1.2$ activity by $\alpha_1AR$–Pyk2–Src signaling as a critical process for control of synaptic strength. Our findings indicate that $Ca_V1.2$ forms a supramolecular signaling complex (signalosome) with PKC, Pyk2, and Src and that $\alpha_1AR$–PKC–Src–$Ca_V1.2$ signaling constitutes a central regulatory mechanism of neuronal activity and synaptic plasticity by NE.

## Results

### $\alpha_1AR$ signaling augments LTCC activity in hippocampal neurons via PKC, Pyk2, and Src

We performed cell-attached recordings from cultured hippocampal neurons for single-channel analysis, which allows pharmacological isolation of LTCCs by application of $\omega$-conotoxins GVIA and MVIIC (*Hall et al., 2013*; *Oliveria et al., 2007*; *Patriarchi et al., 2016*; *Qian et al., 2017*). LTCC channel activity was measured by cell-attached recordings, which yielded the product of the number of channels (N) and the open probability (Po) of each single channel. Application of phenylephrine (PHE), a selective agonist for all three $\alpha_1ARs$, augmented N × Po of LTCCs from 0.18 ± 0.0433 ($H_2O$ vehicle Control, $n = 11$) to 0.6156 ± 0.1386 (PHE; $n = 13$, $p \leq 0.01$; *Figure 1A–C*). This increase was blocked by the selective $\alpha_1AR$ antagonist prazosin (0.2954 ± 0.0607; $n = 10$, $p \leq 0.05$), indicating that PHE acted through $\alpha_1ARs$ and not other G-protein-coupled receptors. Prazosin by itself had no effect, vs. vehicle control (0.1846 ± 0.04624; $n = 10$) suggesting that there is little if any regulation of LTCCs under basal conditions in neurons by $\alpha_1ARs$. PHE also increased the peak current of the ensemble average current in a prazosin-sensitive manner (*Figure 1D, E*).

Because direct phosphorylation of $\alpha_11.2$ by PKC inhibits $Ca_V1.2$ activity in heart (*McHugh et al., 2000*), we explored whether PKC might upregulate $Ca_V1.2$ activity indirectly via other kinases. PKC can activate Pyk2 (*Bartos et al., 2010*; *Dikic et al., 1996*; *Huang et al., 2001*; *Lev et al., 1995*) and thereby Src (*Dikic et al., 1996*; *Huang et al., 2001*). Src, in turn, augments LTCC activity (*Bence-Hanulec et al., 2000*; *Endoh, 2005*; *Gui et al., 2006*; *Hu et al., 1998*; *Strauss et al., 1997*; *Wu et al., 2001*). Therefore, we tested whether block of Pyk2 and Src affects upregulation of LTCC activity by PHE. In a new set of recordings augmentation of LTCC activity by PHE from NPo of 0.2008 ± 0.03348 (dimethyl sulfoxide (DMSO) vehicle control, $n = 33$) to 0.3272 ± 0.04412 (PHE, $n = 33$; $p \leq 0.01$; *Figure 2A–C*) was completely blocked by two different PKC inhibitors, bisindolylmaleimide I (GF109203X; Bis I; 0.1412 ± 0.03305; $n = 11$, $p \leq 0.01$) and chelerythrine (Chel; 0.05801 ± 0.01508; $n = 12$, $p \leq 0.001$), the Pyk2-selective inhibitor PF-719 (0.09118 ± 0.02828; $n = 10$, $p \leq 0.01$) and two structurally different Src family kinase inhibitors, PP2 (0.01487 ± 0.006808; $n = 7$, $p \leq 0.001$) and SU6656 (0.09149 ± 0.02866; $n = 9$, $p \leq 0.01$). Peak currents of ensemble averages showed respective changes (*Figure 2D, E*). Accordingly, $\alpha_1AR$ signaling increases LTCC activity via a PKC–Pyk2–Src signaling cascade. Notably, stimulation of two other major $G_q$-protein-coupled receptors in neurons, that is, the metabotropic mGluR1/5 receptors with dihydroxyphenylglycine (DHPG) and muscarinic receptors with muscarine, did not significantly increase LTCC activity, although there was a tendency for DHPG to do so (*Figure 2—figure supplement 1*).

### PKC augments LTCC activity in hippocampal neurons via Pyk2 and Src

To further establish a role of Pyk2 and Src, we directly stimulated PKC by including PMA in the bath solution during single-channel recording of LTCCs in hippocampal neurons. PMA increased NPo of LTCCs by around twofold from 0.1099 ± 0.0173 (DMSO control, $n = 45$) to 0.232 ± 0.03269 (PMA, $n = 38$; $p \leq 0.001$, *Figure 3A–C*). This increase was blocked by Pyk2 inhibitors PF-719 (NPo = 0.1407 ± 0.02705, $n = 13$, $p \leq 0.05$) and PF-431396 (NPo = 0.09282 ± 0.01765, $n = 18$, $p \leq 0.001$) and by Src inhibitors PP2 (NPo = 0.05614 ± 0.01815, $n = 14$, $p \leq 0.001$) and SU6656 (NPo = 0.02951 ± 0.00555,

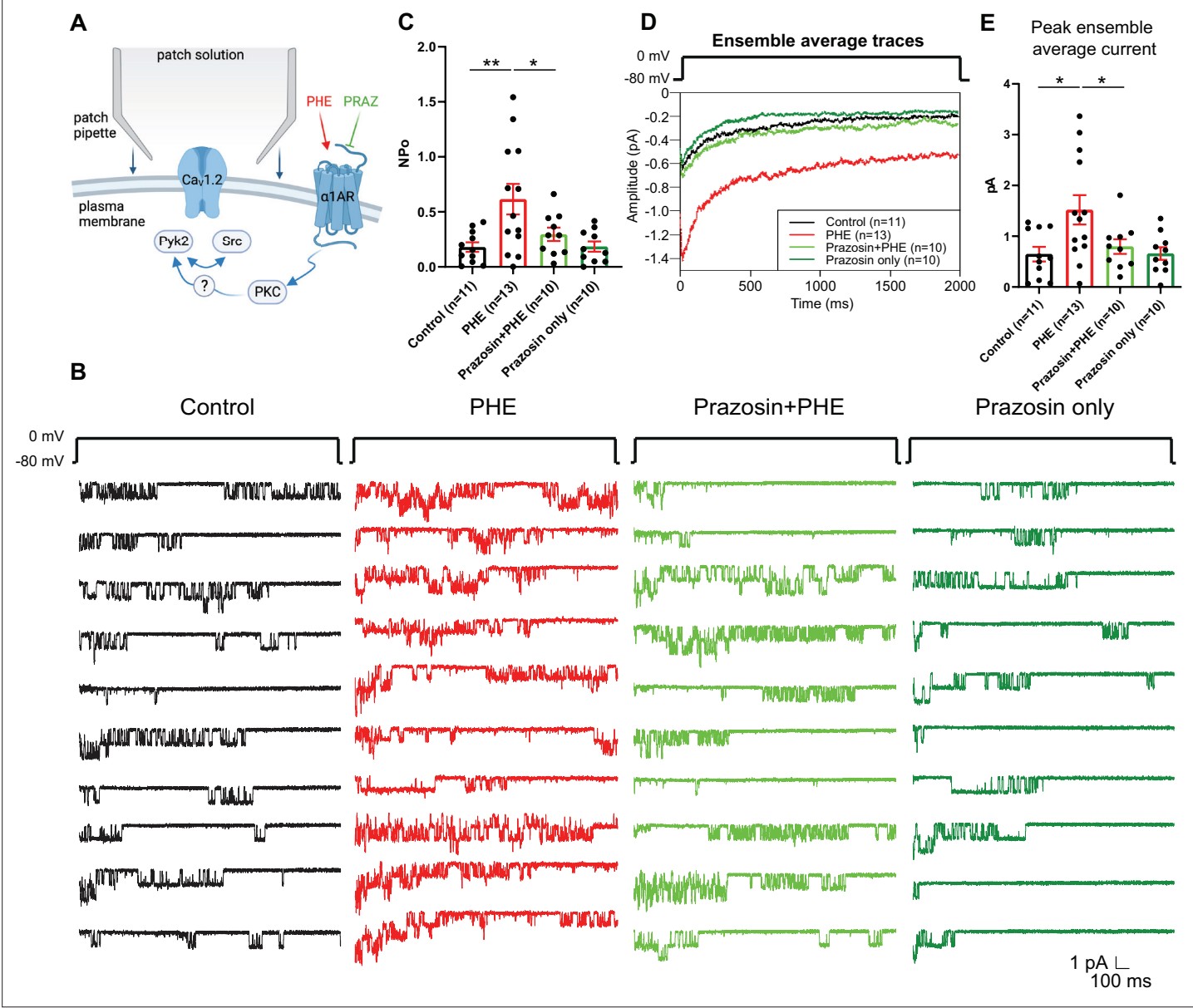

**Figure 1.** The α₁AR agonist phenylephrine (PHE) augments NPo of L-type Ca²⁺ channels (LTCCs) in hippocampal neurons. (**A**) Neurons were preincubated with vehicle, PHE and prazosin (PRAZ) before seal formation. (**B**) Ten consecutive traces from representative cell-attached single-channel recordings of LTCCs from cultured hippocampal neurons with vehicle (water; black), 10 µM PHE (red), PHE plus 20 nM prazosin (bright green), and prazosin alone (dark green). (**C**) The increase in NPo by PHE was blocked by prazosin. $F_{3,40}$ = 5.474. Control vs. PHE, p = 0.0036; PHE vs. Prazosin + PHE, p = 0.0334; Control vs. Prazosin only, p = 0.9723. (**D**) Ensemble averages during depolarization. (**E**) The increase in ensemble average peak currents by PHE was blocked by prazosin. $F_{3,40}$ = 4.506. Control vs. PHE, p = 0.0101; PHE vs. Prazosin + PHE, p = 0.0316; Control vs. Prazosin only, p = 0.9722. (**C, E**) Data are presented as means ± standard error of the mean (SEM). *n* represents the number of cells (*p ≤ 0.05, **p ≤ 0.01; analysis of variance [ANOVA] with post hoc Holm–Sidak's multiple comparisons test). Panel A was created using Biorender.com.

n = 8, p ≤ 0.001). Peak currents of ensemble averages showed respective changes (*Figure 3D, E*). The L-type calcium channel blocker isradipine completely blocked L-type currents in the presence of PMA, indicating successful isolation of L-type single-channel currents (*Figure 3—figure supplement 1*). These results show that in hippocampal neurons, PKC activation stimulates LTCC activity and this augmentation requires Pyk2 and Src activity.

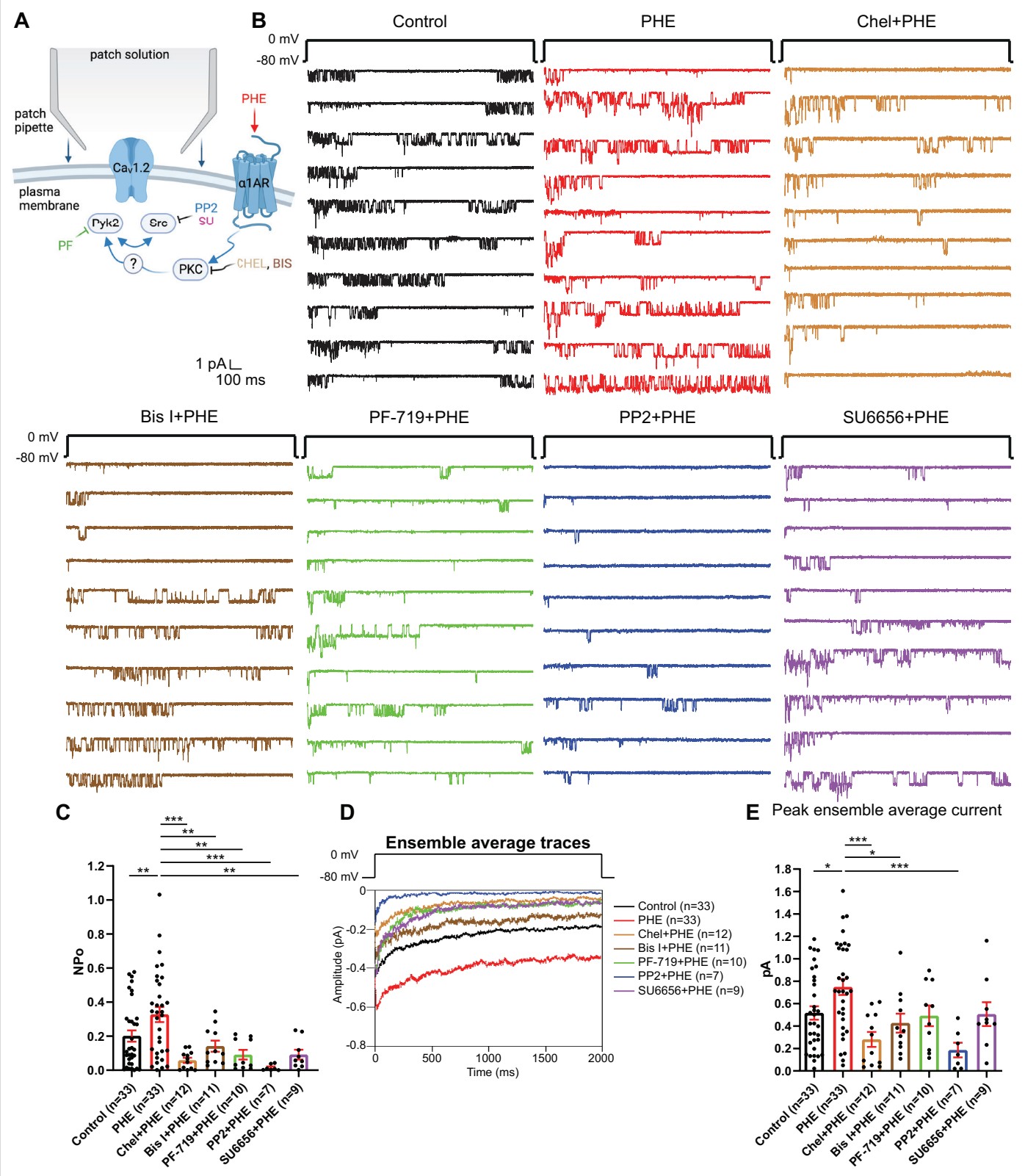

**Figure 2.** The phenylephrine (PHE)-induced increase in NPo of L-type Ca²⁺ channels (LTCCs) in hippocampal neurons requires PKC, Pyk2, and Src. (**A**) Neurons were preincubated with vehicle, PHE, and the indicated kinase inhibitors before seal formation. (**B**) Ten consecutive traces from representative cell-attached single-channel recordings of LTCCs with vehicle (0.1% DMSO; black) and PHE either alone (red) or with the PKC inhibitors chelerythrine (Chel; 10 µM; bright brown) and bisindolylmaleimide I (Bis I; 100 nM; dark brown), the Pyk2 inhibitor PF-719 (1 µM; green), or the Src inhibitors PP2

*Figure 2 continued on next page*

*Figure 2 continued*

(10 µM; blue) and SU6656 (10 µM; purple). (**C**) The increase in NPo by PHE was blocked by all inhibitors. $F_{6,108}$ = 6.434. Control vs. PHE, p = 0.0076; PHE vs. Chel + PHE, p = 0.0001; PHE vs. Bis I + PHE, p = 0.0076; PHE vs. PF-719 + PHE, p = 0.0018; PHE vs. PP2 + PHE, p = 0.0003; PHE vs. SU6656 + PHE, p = 0.0022. (**D**) Ensemble averages during depolarization. (**E**) The increase in ensemble average peak currents by PHE was blocked by PKC inhibitors chelerythrine, bisinolylmaleimide I, and Src inhibitor PP2. $F_{6,108}$ = 4.839. Control vs. PHE, p = 0.0242; PHE vs. Chel + PHE, p = 0.0004; PHE vs. Bis I + PHE, p = 0.0242; PHE vs. PF-719 + PHE, p = 0.0723; PHE vs. PP2 + PHE, p = 0.0006; PHE vs. SU6656 + PHE, p = 0.0723. (**C, E**) Data are presented as means ± standard error of the mean (SEM). *n* represents the number of cells (*p ≤ 0.05, **p ≤ 0.01, ***p ≤ 0.001; analysis of variance [ANOVA] with post hoc Holm–Sidak's multiple comparisons test). Panel A was created using Biorender.com.

The online version of this article includes the following figure supplement(s) for figure 2:

**Figure supplement 1.** Group I mGluR and muscarinic receptor agonists did not change NPo of L-type $Ca^{2+}$ channels (LTCCs) in hippocampal neurons.

## $\alpha_1$AR signaling augments single-channel open probability Po of LTCCs in neurons

Preincubation of neurons with PHE could promote either surface insertion or Po of LTCCs. To test whether PHE augmented specifically Po, we used pipettes with smaller diameters to minimize patch size and channel number in the patch, as reflected by pipette resistences of 7–12 vs. 3.5–5.5 MΩ in the preceding experiments. This approach typically resulted in <4 channels per patch, allowing exact determination of channel number and thereby calculation of single-channel Po. PHE was acutely washed on after establishing baseline activity to avoid delays as occurring when recording the effect of preincubation of neurons with PHE during which new channels could have been inserted (*Figure 4A*). PHE consistently increased within 2–3 min Po and peak currents as determined by ensemble averages (*Figure 4B–D*).

Application of the endogenous agonist NE to the outside of the cell-attached pipette was equally able to augment single-channel Po and peak currents of ensemble averages (*Figure 4E–G*). Of note, the $\beta_2$AR-selective adrenergic agonist albuterol can also augment Po of $Ca_V$1.2 by stimulating the $Ca_V$1.2-associated $\beta_2$AR, adenylyl cyclase and PKA (*Davare et al., 2001*; *Dittmer et al., 2014*; *Murphy et al., 2014*; *Oliveria et al., 2007*; *Patriarchi et al., 2016*; *Qian et al., 2017*). However, it does so only when applied inside the patch pipette and not when applied after seal formation to the outside, reflective of highly localized, spatially restricted signaling events (*Davare et al., 2001*; *Dittmer et al., 2014*; *Murphy et al., 2014*; *Oliveria et al., 2007*; *Patriarchi et al., 2016*; *Qian et al., 2017*). Accordingly, NE applied to the outside of the pipette augments Po not via $\beta_2$AR but rather via $\alpha_1$AR signaling. Consistently, the increases in single-channel Po and peak currents of ensemble averages seen with NE were remarkably similar to the respective PHE effects.

To further test the role of $\alpha_1$AR vs. $\beta_2$AR signaling in this recording configuration, we applied NE either alone or together with the $\alpha_1$AR antagonist prazosin to the neurons before seal formation and recording of channel activity (*Figure 5A*). NE increased Po and peak current of ensemble averages more strongly in this approach than when applied only to the outside of the pipettes (*Figure 5B–D*). This effect was inhibited but not fully blocked when prazosin was co-applied with NE. These two effects are consistent with upregulation of $Ca_V$1.2 activity by NE via both $\alpha_1$AR and $\beta_2$AR signaling.

## BK augments Po of LTCCs in neurons

The above results indicate that signaling by the $G_q$-coupled $\alpha_1$AR promotes LTCC activity in a manner that is spatially much less localized if at all as opposed to signaling by the $G_s$-coupled $\beta_2$AR. Stimulation of other prominent $G_q$-coupled receptors, mGluR and muscarinic receptors, yielded little or no effects, respectively, on LTCC activity (*Figure 2—figure supplement 1*), possibly because those might be to far removed from the LTCCs in soma where the recordings were performed. Another $G_q$-coupled receptor that is prominent in the hippocampus is the BK receptor 2 (BK2). Application of BK to the outside of the patch pipette after establishing baseline activity of LTCCs (*Figure 6A*) significantly augmented Po (*Figure 6B–D*). In this set of experiments, the identity of the $Ca^{2+}$ channels in the patch was confirmed by applying the LTCC activity promoting Bay K8644, which, consistently, augmented the current under the patch.

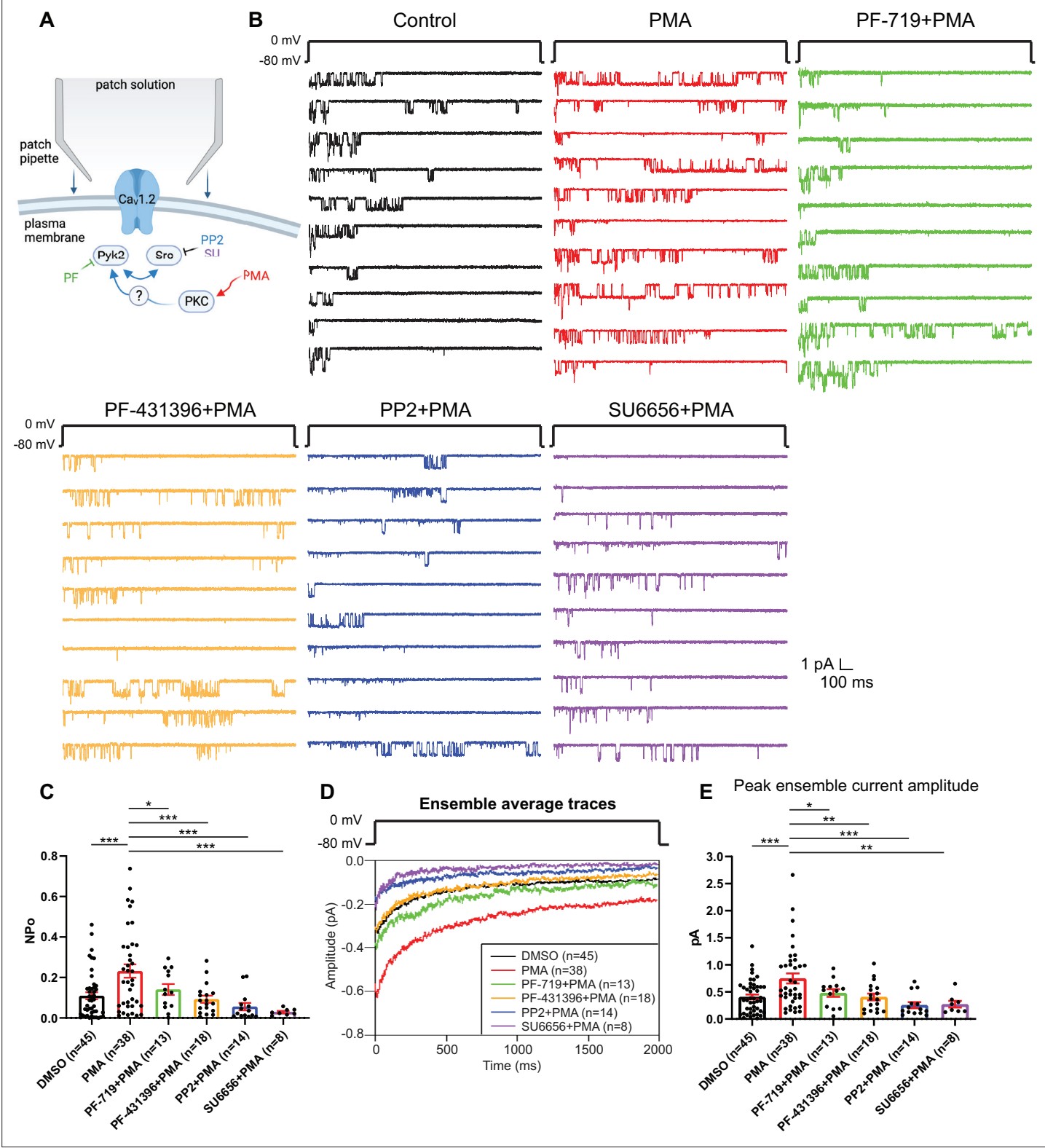

**Figure 3.** The increase in NPo of L-type Ca$^{2+}$ channels (LTCCs) in hippocampal neurons by PKC requires Pyk2 and Src. (**A**) Neurons were preincubated with vehicle, the phorbol ester phorbol-12-myristate-13-acetate (PMA), and the indicated kinase inhibitors before seal formation. (**B**) Ten consecutive traces from representative cell-attached single-channel recordings of LTCCs with vehicle (0.06% DMSO; black) and 2 µM PMA either alone (red) or with the Pyk2 inhibitors PF-719 (1 µM; green) and PF-431396 (3 µM; orange), or the Src inhibitors PP2 (10 µM; blue) and SU6656 (10 µM; purple). (**C**) The increase in NPo by PMA was blocked by all inhibitors. $F_{5,130}$ = 6.530. DMSO vs. PMA, p = 0.0003; PMA vs. PF-719 + PMA, p = 0.0372, PMA vs. PF-431396

*Figure 3 continued on next page*

*Figure 3 continued*

+ PMA, p = 0.0009; PMA vs. PP2 + PMA, p = 0.0003; PMA vs. SU6656 + PMA, p = 0.0005. (**D**) Ensemble averages during depolarization. (**E**) The increase in ensemble average peak currents by PMA was blocked by all inhibitors. $F_{5,130}$ = 5.665. DMSO vs. PMA, p = 0.0003; PMA vs. PF-719 + PMA, p = 0.0303, PMA vs. PF-431396 + PMA, p = 0.0051; PMA vs. PP2 + PMA, p = 0.0003; PMA vs. SU6656 + PMA, p = 0.0051. (**C, E**) Data are presented as means ± standard error of the mean (SEM). *n* represents the number of cells (*p ≤ 0.05, **p ≤ 0.01, ***p ≤ 0.001; analysis of variance [ANOVA] with post hoc Holm–Sidak's multiple comparisons test). Panel A was created using Biorender.com.

The online version of this article includes the following figure supplement(s) for figure 3:

**Figure supplement 1.** L-type channel blocker isradipine completely blocks L-type single-channel currents in the presence of phorbol-12-myristate-13-acetate (PMA).

## Pyk2 co-immunoprecipitates with Ca$_V$1.2 from brain and heart

Kinases and proteins that regulate kinase activity are often found in complexes with their ultimate target proteins (i.e., their ultimate substrates) including different ion channels for efficient and specific signaling (*Dai et al., 2009*; *Dodge-Kafka et al., 2006*). Both, PKC (*Navedo et al., 2008*; *Yang et al., 2005*) and Src (*Bence-Hanulec et al., 2000*; *Chao et al., 2011*; *Hu et al., 1998*), are associated with Ca$_V$1.2. We tested in brain and heart (where Ca$_V$1.2 is most abundant) whether the same is true for

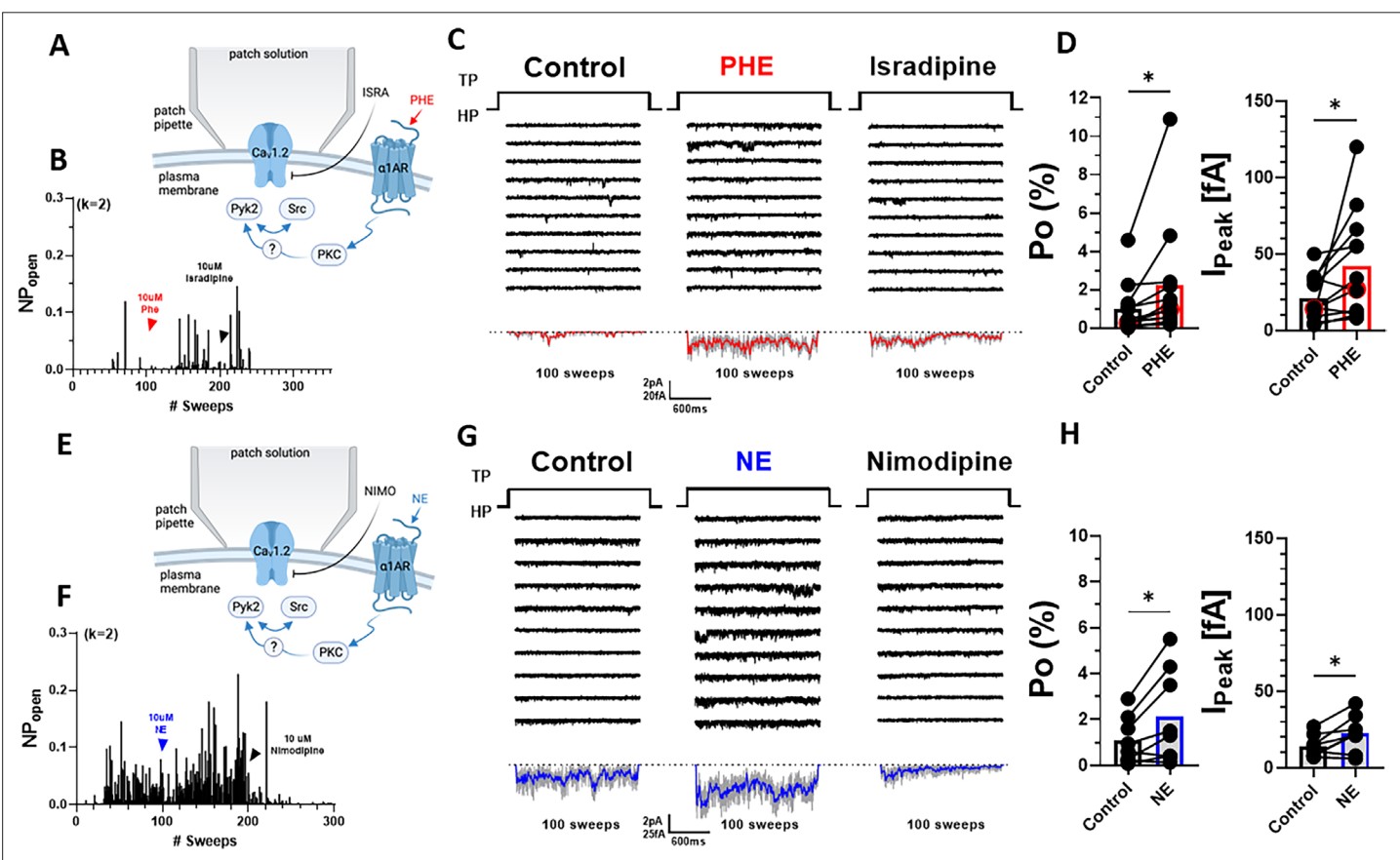

**Figure 4.** α$_1$AR signaling augments Po of L-type Ca$^{2+}$ channels (LTCCs) in hippocampal neurons. (**A, E**) Seals were formed by the recording pipettes before application of phenylephrine (PHE) or norepinephrine (NE) and ultimately of either isradipine or nimodipine to ensure channel activity was mediated by LTCCs. (**B**) Sample diary shows time course of Po before and after application of 10 μM PHE and then 10 μM isradipine. The number of channels under the patch was estimated based on the maximal number of observed stagged openings in each patch (*k*; upper left). (**C**) Ten consecutive traces from representative cell-attached single-channel LTTC recordings before and after application of PHE and then isradipine. Bottom panels show ensemble averages. (**D**) PHE increases Po (left) and peak currents of ensemble averages (*n* = 12 cells; right). (**F**) Sample diary shows time course of Po before and after application of 10 μM NE and then 10 μM isradipine. (**G**) Ten consecutive traces from representative cell-attached single-channel recordings of LTCCs before and after application of NE and then nimodipine. Bottom panels show ensemble averages. (**H**) NE increases Po (left) and peak currents of ensemble averages (*n* = 8 cells; right). (**D, H**) Data are presented as means ± standard error of the mean (SEM). Statistical significance was tested by a paired, two-tailed Student's *t*-test, *p ≤ 0.05. Panels A and E were created using Biorender.com.

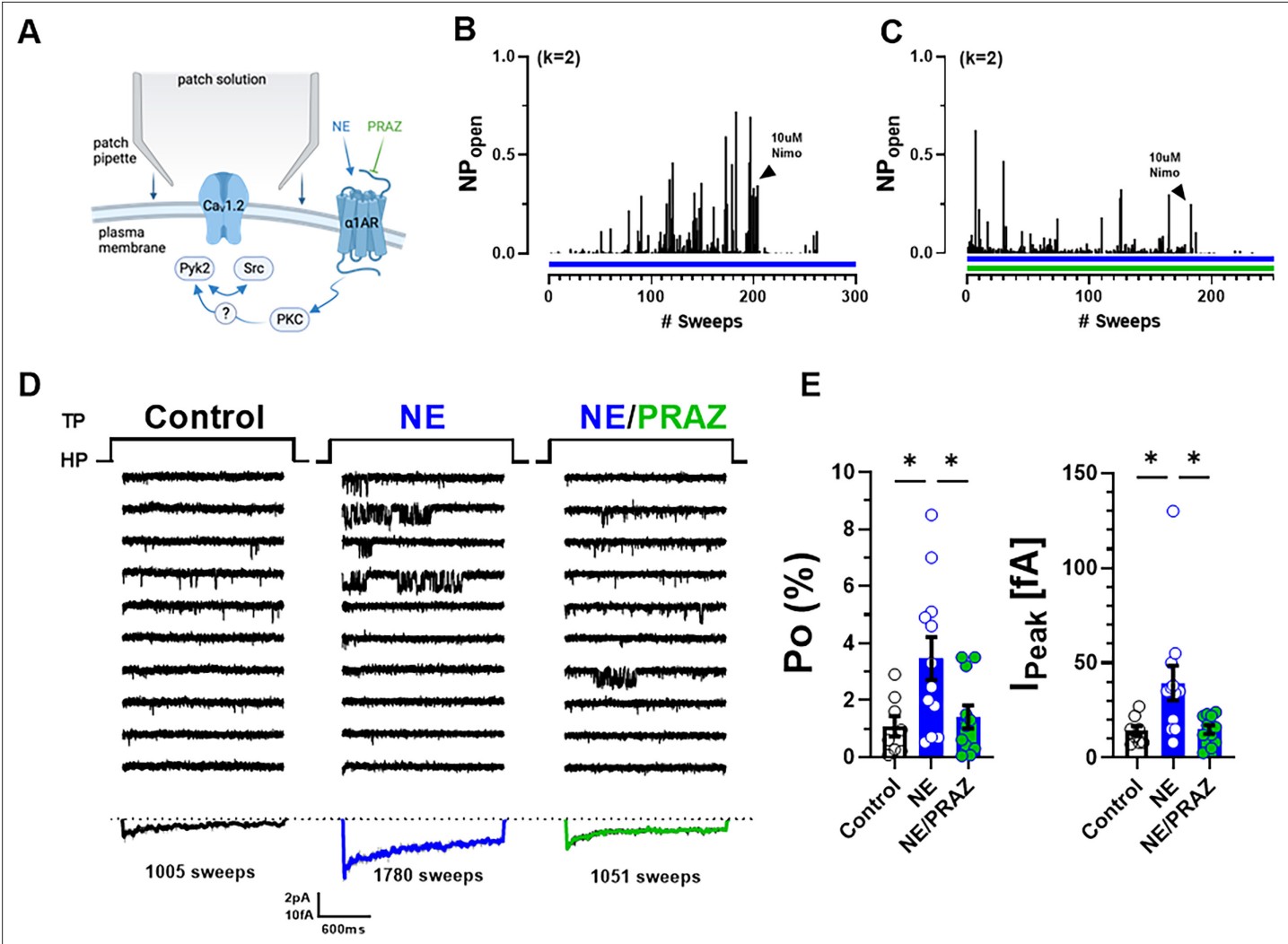

**Figure 5.** Norepinephrine (NE) can augment Po of L-type Ca²⁺ channels (LTCCs) via $\alpha_1$AR signaling in hippocampal neurons. (**A**) Neurons were preincubated with NE ± prazosin (PRAZ) before seal formation. (**B, C**) Sample diaries show time courses of Po recordings obtained after preincubation with either NE alone or NE + PRAZ and seal formation. The number of channels under the patch was estimated based on the maximal number of observed stagged openings in each patch ($k$; upper left). (**D**) Ten consecutive traces from representative cell-attached single-channel recordings of LTCCs under control conditions or upon pre-incubation with either NE alone or NE plus PRAZ. Bottom panels show ensemble averages. (**E**) NE strongly increases Po (left) and peak currents of ensemble averages (right), which was strongly but not fully inhibited by PRAZ. Data are presented as means ± standard error of the mean (SEM; Control, $n = 8$ cells; NE, $n = 12$ cells; NE/PRAZ, $n = 11$ cells). Statistical significance was tested by a one-way analysis of variance (ANOVA) with Bonferroni correction, *p ≤ 0.05. Panel A was created using Biorender.com.

Pyk2. The Pyk2 antibody detected a single immunoreactive band with an apparent $M_R$ of ~120 kDa in brain lysate (*Figure 7A*) and two bands in the same range in heart (*Figure 7A, B*), as reported earlier (*Dikic et al., 1998*). The shorter form is missing 42 residues in the proline-rich region of Pyk2, which affects its binding selectivity to proteins with SH3 domains. The single size form of Pyk2 present in brain and its two size forms expressed in heart co-immunoprecipitated with $Ca_V1.2$ (*Figure 7A, B*). No Pyk2 immunoreactive band was detectable when the immunoprecipitation (IP) was performed with control IgG, demonstrating that the co-IP of Pyk2 with $Ca_V1.2$ was specific. The detergent extracts from brain and heart were cleared of non-soluble material by ultracentrifugation prior to co-IP of Pyk2 with $Ca_V1.2$. Thus, our findings indicate that Pyk2 forms a bona fide protein complex with $Ca_V1.2$ rather than just co-residing in a detergent-resistant subcellular compartment. We also confirmed earlier work (*Figure 7A*, bottom panel) that indicated association of Src with $Ca_V1.2$ in vitro (*Bence-Hanulec et al., 2000*; *Endoh, 2005*; *Gui et al., 2006*; *Hu et al., 1998*; *Strauss et al., 1997*; *Wu et al., 2001*) and in intact cells (*Bence-Hanulec et al., 2000*; *Chao et al., 2011*; *Hu et al., 1998*).

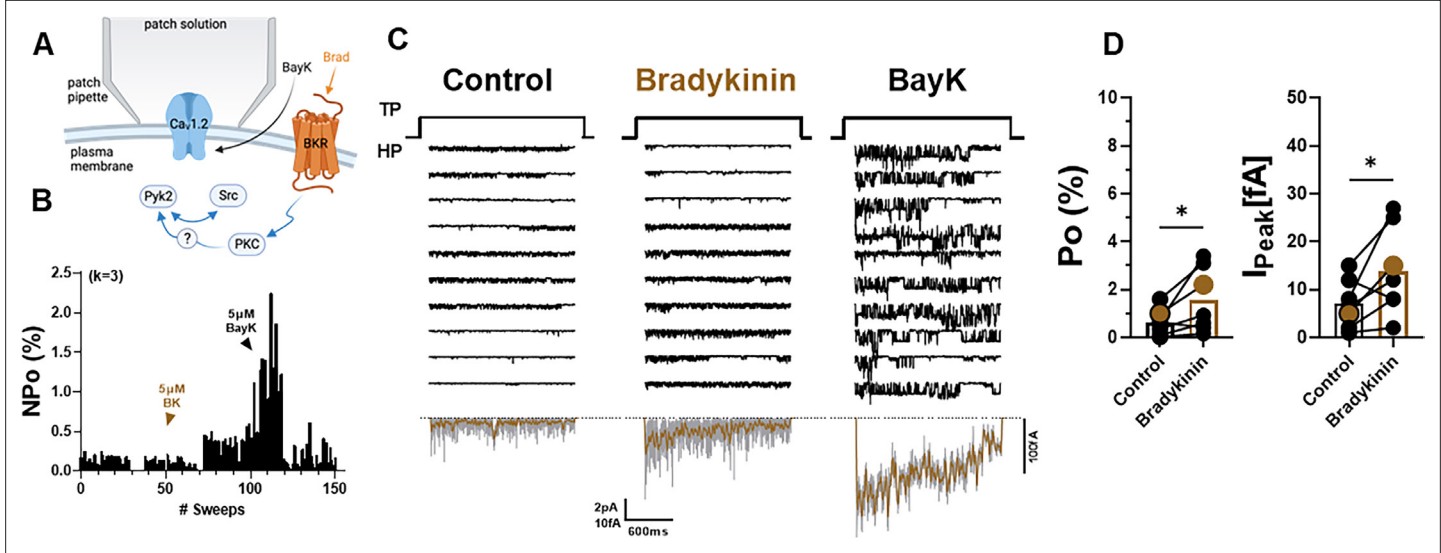

**Figure 6.** Bradykinin (BK) signaling augments Po of L-type Ca²⁺ channels (LTCCs) in hippocampal neurons. (**A**) Seals were formed by the recording pipettes before application of BK and ultimately of BayK8644 (BayK) to ensure channel activity was mediated by LTCCs. (**B**) Sample diary shows time course of Po before and after application of 5 µM BK and then 5 µM BayK to not only provide further evidence that the channels in the patch were LTCC but also aid in determining channel number $k$ (upper left), which is the number of channels under the patch as estimated based on the maximal number of observed stagged openings in each patch. (**C**) Ten consecutive traces from representative cell-attached single-channel recordings of LTCCs before and after application of BK and then BayK. Bottom panels show ensemble averages. (**D**) BK increases Po (left) and peak currents of ensemble averages (right). Data are presented as means ± standard error of the mean (SEM; $n$ = 7 cells). Statistical significance was tested by a paired, two-tailed Students $t$-test, *p ≤ 0.05. Panel A was created using Biorender.com.

## Pyk2 binds to the loop between domains II and III of $\alpha_1$1.2

To further confirm a direct interaction between Pyk2 and Ca$_V$1.2 we performed pulldown experiments using purified solubilized His-tagged Pyk2 and purified bead-bound GST-tagged $\alpha_1$1.2 fragments covering all intracellular regions of $\alpha_1$1.2 (*Supplementary file 1*, *Snutch et al., 1990*). As demonstrated earlier, all $\alpha_1$1.2 fragments were present in comparable amounts (*Hall et al., 2013*; *Patriarchi et al., 2016*). The GST fusion protein covering the loop between domains II and III of $\alpha_1$1.2 specifically pulled down Pyk2 (*Figure 7C, D*) indicating that Pyk2 directly binds to this region of the $\alpha_1$ subunit.

## Inhibitors of Pyk2 and Src block the increase in $\alpha_1$1.2 tyrosine phosphorylation upon stimulation of PKC

PC12 cells are of neural-endocrine crest origin and widely used as model cells for neuronal signaling and development. They express high levels of Ca$_V$1.2 (*Eiki et al., 2009*; *Mustafa et al., 2010*; *Taylor et al., 2000*; *Walter et al., 2000*), the BK receptor, and Pyk2 (*Bartos et al., 2010*; *Dikic et al., 1996*; *Lev et al., 1995*), making them an ideal model system for the difficult biochemical analysis of Ca$_V$1.2 phosphorylation. To characterize tyrosine phosphorylation of $\alpha_1$1.2 we performed IP with the general anti-phosphotyrosine antibody 4G10 (*Clifton et al., 2004*; *Ward et al., 1992*). For this purpose, lysates were extracted with 1% sodium dodecyl sulfate (SDS) at 65°C followed by neutralization of SDS and ultracentrifugation before IP with the general anti-phosphotyrosine antibody 4G10 (*Clifton et al., 2004*; *Ward et al., 1992*). IP with 4G10 followed by immunoblotting (IB) with antibodies against the protein of interest is more reliable and more broadly applicable than the inverse. Because $\alpha_1$1.2 does not re-associate with its binding partners after complex dissociation with SDS and the neutralization and dilution of SDS with Triton X-100 (*Davare et al., 1999*; *Hell et al., 1995*; *Hell et al., 1993b*; see also *Leonard and Hell, 1997*), detection of $\alpha_1$1.2 by IB in the 4G10 IP would reflect specific tyrosine phosphorylation of the $\alpha_1$1.2 subunit and not its artefactual re-association with an $\alpha_1$1.2-associating tyrosine-phosphorylated protein that had been pulled down by the 4G10 antibody. This approach also allows analysis of tyrosine phosphorylation of Pyk2 within the same sample.

PC12 cells were pretreated with vehicle (0.02% DMSO), the Pyk2 inhibitor PF-431396, or the Src inhibitors SU6656 and PP2 or its inactive analogue PP3 for 5 min before application of BK or PMA for

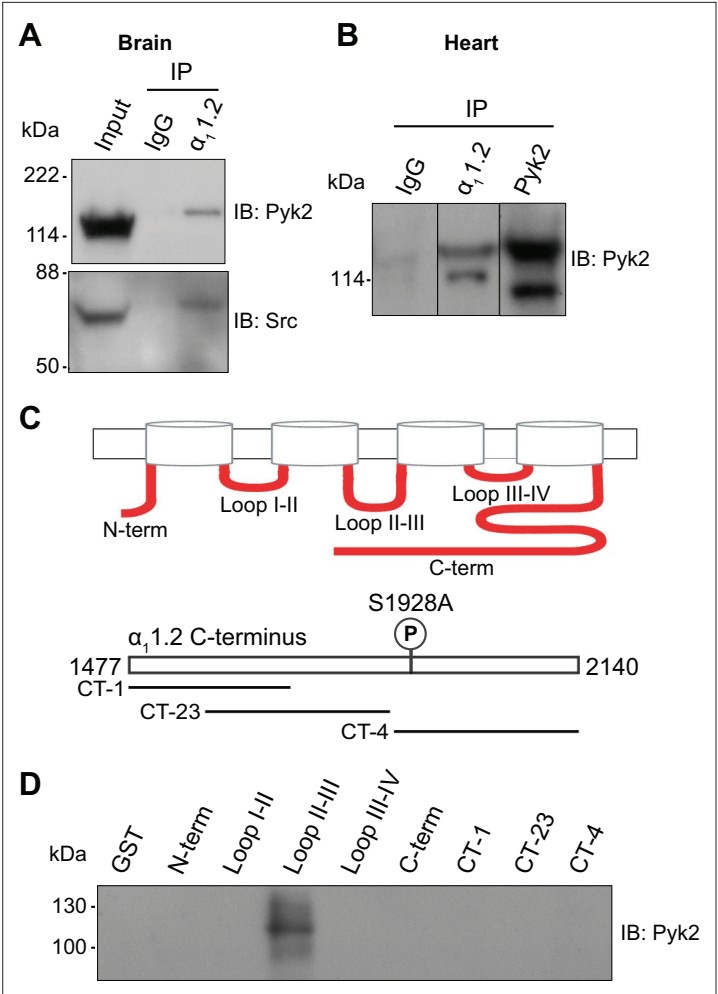

**Figure 7.** Pyk2 binds to the loop between domains II and III of $\alpha_1 1.2$. Co-immunoprecipitation of Pyk2 and Src with Ca$_V$1.2 from brain (**A**) and heart (**B**). Triton X-100 extracts were cleared from non-soluble material by ultracentrifugation before immunoprecipitation (IP) with antibodies against $\alpha_1 1.2$, Pyk2 itself, or non-immune control antibodies (rabbit IgG) and immunoblotting (IB) with anti-Pyk2 and anti-Src. Brain lysate (**A**, Input; 20 µl) and Pyk2 immunoprecipitates (**B**) served as positive control for detection of Pyk2 and Src by IB. Lanes for rabbit IgG control and $\alpha_1 1.2$ IP in B are from the same IB as the Pyk2 IP, which is depicted from a shorter exposure than the IgG and $\alpha_1 1.2$ IP lanes because IB signal was much stronger after Pyk2 IP than $\alpha_1 1.2$ IP. Comparable results were obtained in four independent experiments. (**C**) Schematic diagram of the intracellular $\alpha_1 1.2$ fragments used in the pulldown assay (***Supplementary file 1***). (**D**) Pulldown assay of Pyk2 binding to $\alpha_1 1.2$ fragments. GST fusion proteins of the N-terminus, the loops between domains I and II, II and III, III and IV, the whole C-terminus, and three different overlapping fragments covering the C-terminus were expressed in *Escherichia coli*, immobilized on glutathione Sepharose, washed and incubated with purified His-tagged Pyk2. Comparable amounts of fusion proteins were present (data not shown but see ***Hall et al., 2013***; ***Hall et al., 2007***; ***Patriarchi et al., 2016***; ***Xu et al., 2010***). Comparable results were obtained in five independent experiments.

The online version of this article includes the following source data for figure 7:

**Source data 1.** Original files of the full raw unedited blots with bands labeled in red boxes.

10 min. BK strongly activates Pyk2 in PC12 cells via its G$_q$-coupled cognate receptor (***Dikic et al., 1996***; ***Lev et al., 1995***). Using the 4G10 IP method, we found that both PMA and BK increased tyrosine phosphorylation of Pyk2 as previously described (***Dikic et al., 1996***; ***Lev et al., 1995***; ***Figure 8A–C***). This increase was prevented by PF-431396. In parallel, we determined phosphorylation of Pyk2 on Y402 and Y579 by direct IB of PC12 lysates with corresponding phosphospecific antibodies. Upon stimulation via PKC, Pyk2 phosphorylates itself in trans on Y402 (***Bartos et al., 2010***; ***Park et al., 2004***) and then binds with phosphoY402 to the SH2 domain of Src (***Dikic et al., 1996***). This binding

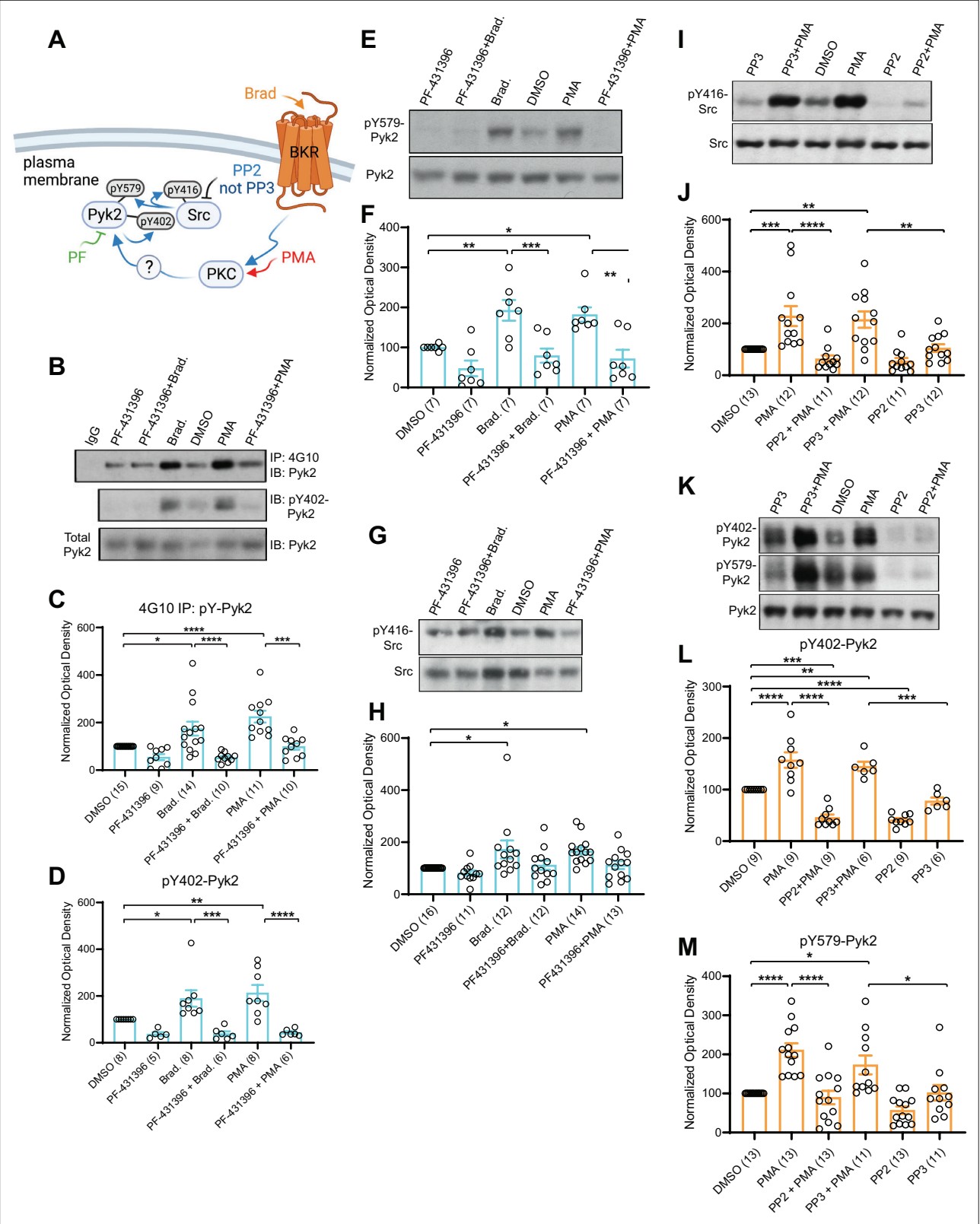

**Figure 8.** PKC activates interdependent Pyk2 and Src. PC12 cells were pretreated with vehicle (0.02% DMSO), the Pyk2 inhibitor PF-431396 (3 µM), and the Src inhibitor PP2 (10 µM) or its inactive analogue PP3 (10 µM) for 5 min before application of bradykinin (Brad., 2 µM) or phorbol-12-myristate-13-acetate (PMA, 2 µM) for 10 min, extraction with 1% sodium dodecyl sulfate (SDS) at 65°C to ensure dissociation of all proteins, neutralization of SDS with excess of Triton X-100, and ultracentrifugation. Supernatants were analysed by direct immunoblotting (IB) with the indicated Pyk2 and Src antibodies.

*Figure 8 continued on next page*

*Figure 8 continued*

Some samples underwent immunoprecipitation (IP) with the anti-phosphotyrosine antibody 4G10 before IB with anti-Pyk2 antibody (top panel in **A** and quantification in **B**). IgG indicates control IP with non-immune mouse IgG. (**A**) Schematic diagram depicting the bradykinin receptor–PKC–Pyk2/Src signaling cascade and drugs used to target each molecular entity. (**B**) Upper panel: Total pY levels of Pyk2 determined by IP with 4G10 and IB with anti-Pyk2. Middle panel: pY402 levels of Pyk2 detected with anti-pY402 in corresponding lysates. Lower panel: Levels of total Pyk2 detected with anti-Pyk2 in same lysates. (**C**) Ratios of total pY of Pyk2 after 4G10 IP to total Pyk2 in lysates, normalized to control. $F_{5,63}$ = 12.73. DMSO vs. Brad., p = 0.012; DMSO vs. PMA, p < 0.0001; Brad. vs. PF-431396 + Brad., p < 0.0001; PMA vs. PF-431396 + PMA, p = 0.0001. (**D**) Ratios of pY402 to total Pyk2 signals in lysates, normalized to control. $F_{5,35}$ = 10.94. DMSO vs. Brad., p = 0.039; DMSO vs. PMA, p = 0.0052; Brad. vs. PF-431396 + Brad., p = 0.0005; PMA vs. PF-431396 + PMA, p < 0.0001. (**E**) Upper panel: pY579 levels of Pyk2 detected with anti-pY579. Lower panel: Levels of total Pyk2 detected with anti-Pyk2 in same lysates. (**F**) Ratios of pY579 to total Pyk2 signals in lysates, normalized to control. $F_{5,36}$ = 10.18. DMSO vs. Brad., p = 0.0072; DMSO vs. PMA, p = 0.021; Brad. vs. PF-431396 + Brad., p = 0.0008; PMA vs. PF-431396 + PMA, p = 0.0011. (**G, I**) Upper panels: pY416 levels of Src detected with anti-pY416. Lower panels: Levels of total Src detected with anti-Src in same lysates. (**H, J**) Ratios of pY416 to total Src signals in lysates, normalized to control. (**H**) $F_{5,72}$ = 4.464. DMSO vs. Brad., p = 0.0167; DMSO vs. PMA, p = 0.0226. (**J**) $F_{5,65}$ = 11.06. DMSO vs. PMA, p = 0.001; PMA vs. PP2 + PMA, p < 0.0001; DMSO vs. PP3 + PMA, p = 0.0042; PP3 vs. PP3 + PMA, p = 0.0086. (**K**) Upper panel: pY402 levels of Pyk2 detected with anti-pY402. Middle panel: pY579 levels of Pyk2 detected with anti-pY579. Lower panel: Levels of total Pyk2 detected with anti-Pyk2 in same lysates. (**L, M**) Ratios of pY402 and pY579 to total Pyk2 signals in lysates, normalized to control. (**L**) $F_{5,42}$ = 35.85. DMSO vs. PMA, p < 0.0001; PMA vs. PP2 + PMA, p < 0.0001; PP3 vs. PP3 + PMA, p = 0.0001; DMSO vs. PP3 + PMA, p = 0.0068; DMSO vs. PP2 + PMA, p = 0.0001; DMSO vs. PP2, p < 0.0001. (**M**) $F_{5,68}$ = 13.40. DMSO vs. PMA, p < 0.0001; PMA vs. PP2 + PMA, p < 0.0001; PP3 vs. PP3 + PMA, p = 0.0362; DMSO vs. PP3 + PMA, p = 0.0202. (**C, D, F, H, J, L, M**) Data are presented as mean ± standard error of the mean (SEM). Number (*n*) of independent experiments for each condition are indicated inside bars. Statistical analysis was by analysis of variance (ANOVA) with post hoc Bonferroni's multiple comparisons test; *p ≤ 0.05, **p ≤ 0.01, ***p ≤ 0.001, ****p ≤ 0.0001. Bradykinin- and PMA-induced phosphorylation of Pyk2 on Y402 and Y579 and of Src on Y416, all of which were blocked by PF-431396 and PP2 but not the inactive PP3. Panel A was created using Biorender.com.

The online version of this article includes the following source data for figure 8:

**Source data 1.** Original files of the full raw unedited blots with bands labeled in red boxes.

stimulates Src (*Dikic et al., 1996*), which in turn phosphorylates Pyk2 on Y579 in its activation loop for full activation (*Avraham et al., 2000*; *Figure 8A*). Src also phosphorylates itself in trans on Y416 in its activation loop for its own full activation (*Roskoski, 2015*), which was determined in parallel with a phosphospecific antibody. We found that BK and PMA increased phosphorylation of Pyk2 on Y402 (*Figure 8B–D*) and Y579 (*Figure 8E, F*) and of Src on Y416 (*Figure 8G, H*). PF-431396 blocked all of these phosphorylations indicating that Pyk2 acts downstream of PKC and upstream of Src. Furthermore, the Src inhibitor PP2, but not its inactive analog, PP3, also prevented PMA-induced Src autophosphorylation on Y416 (*Figure 8I, J*), as expected. Finally, PP2 inhibited PMA-induced phosphorylation of Pyk2 on Y402 and Y579 indicative of a self-maintaining positive feedback loop between Pyk2 and Src (*Figure 8K–M*). These results support the specific activation of both, Pyk2 and Src under our conditions and suggest that this activation occurs in a self-sustaining manner, which creates a quasi-molecular memory (*Figure 8A*).

Importantly, PMA and BK induced tyrosine phosphorylation of $\alpha_1 1.2$ (*Figure 9*). This effect was blocked by the Pyk2 inhibitor PF-431396 (*Figure 9A–C*) and the Src inhibitors SU6656 and PP2, whereas the inactive PP2 analogue PP3 was without effect (*Figure 9D, E*). These data show that activation of PKC translates into tyrosine phosphorylation of $\alpha_1 1.2$ and that this requires both Pyk2 and Src.

## Knock down of Pyk2 and Src prevents the increase in $\alpha_1 1.2$ tyrosine phosphorylation upon stimulation of PKC

To control for any potential side effects of PF-431396 and determine whether Pyk2 is required for PKC-induced tyrosine phosphorylation of $\alpha_1 1.2$ we employed FIV and HIV lentiviral expression vectors for shRNAs targeting Pyk2 in PC12 cells. We first designed and cloned an shRNA-targeting rat Pyk2 (Sh1) into the FIV-based plasmid pVETL-GFP (*Bartos et al., 2010*; *Boudreau and Davidson, 2012*; *Harper et al., 2006*) and tested the ability and specificity of this construct to knockdown ectopically expressed Pyk2 in HEK293T/17 cells. Cells were co-transfected with vectors for expression of GFP-tagged rat Pyk2 (rPyk2-GFP) and the pVETL-Sh1-GFP or no shRNA control pVETL-GFP (*Figure 10A*) and Pyk2 expression levels in the transfected cell lysates were examined via IB. Expression of rPyk2-GFP was virtually abolished by pVETL-Sh1-GFP whereas the control pVETL-GFP had no effect (*Figure 10A*). IB with both tubulin and GAPDH antibodies confirmed that total protein levels were not affected by transfection of these plasmids (*Figure 10A*).

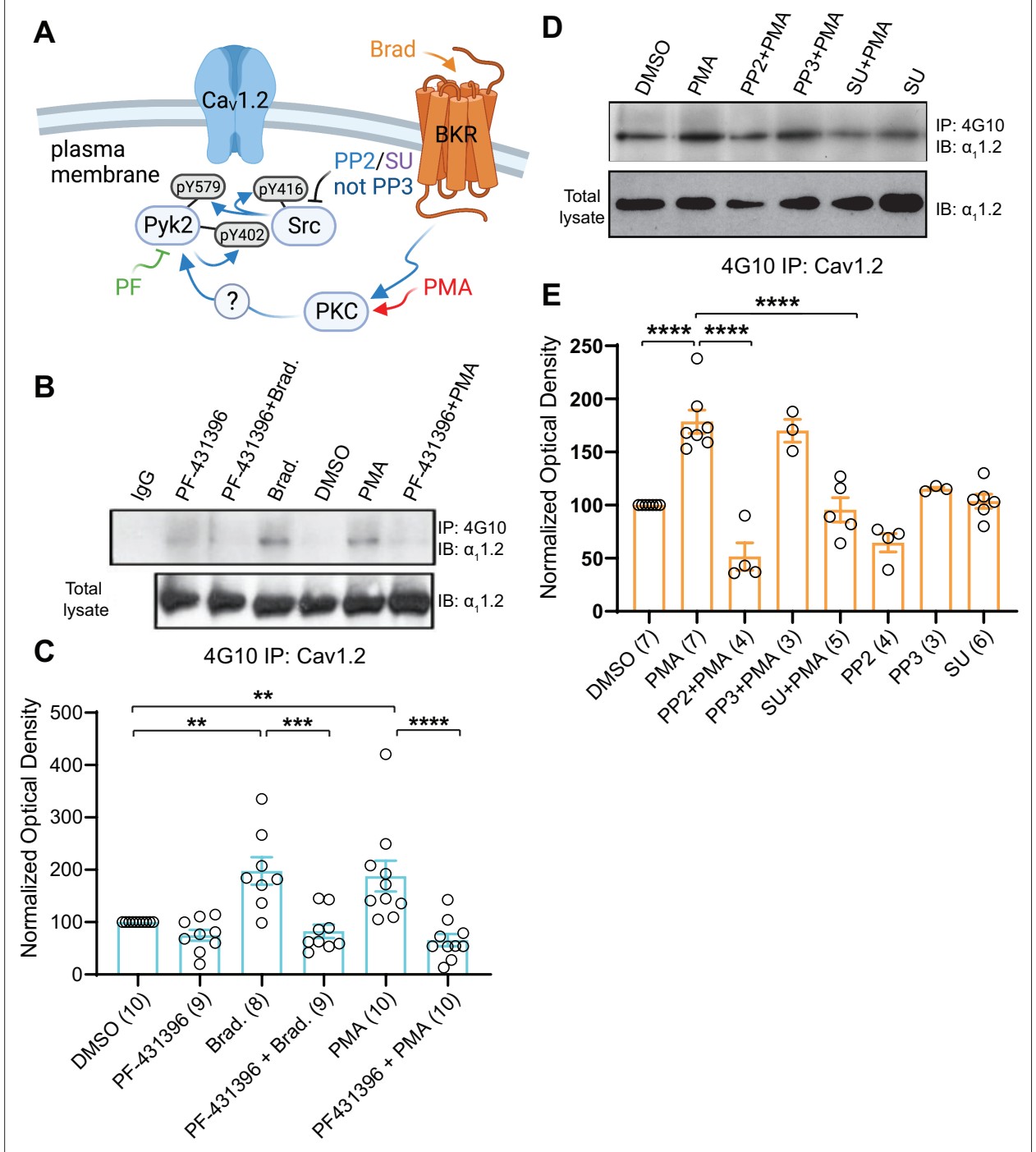

**Figure 9.** Increase in $\alpha_1 1.2$ tyrosine phosphorylation by PKC is blocked by inhibitors or Pyk2 and Src. PC12 cells were treated as in *Figure 8* for analysis of tyrosine phosphorylation by immunoprecipitation (IP) with 4G10 and immunoblotting (IB) with anti-$\alpha_1 1.2$. IgG indicates control IP with non-immune mouse IgG. Vehicle (0.02% DMSO), PF-431396 (3 µM), PP2 (10 µM), PP3 (10 µM), or SU6656 (SU, 10 µM) were applied 5 min before phorbol-12-myristate-13-acetate (PMA) or bradykinin (Brad.) when indicated. (**A**) Schematic diagram depicting the bradykinin receptor–PKC–Pyk2/Src–Ca$_V$1.2 signaling cascade and drugs used to target each molecular entity. (**B, D**) Upper panels: pY of $\alpha_1 1.2$ determined by 4G10 IP and $\alpha_1 1.2$ IB. Lower panels: Levels of total $\alpha_1 1.2$ detected with anti-$\alpha_1 1.2$ in corresponding lysates. (**C, E**) Ratios of pY signals in 4G10 IPs by IB with anti-$\alpha_1 1.2$ to $\alpha_1 1.2$ signals in lysates, normalized to control. Data are presented as mean ± standard error of the mean (SEM). Number (*n*) of independent experiments for each condition are indicated inside bars. Statistical analysis was by analysis of variance (ANOVA) with post hoc Bonferroni's multiple comparisons test. (**C**) $F_{5,50} = 10.65$. DMSO vs. Brad., p = 0.0021; DMSO vs. PMA, p = 0.0036; Brad. vs. PF-431396 + Brad., p = 0.0003; PMA vs. PF-431396 + PMA, p < 0.0001. (**E**) $F_{7,31} = 23.67$. DMSO vs. PMA, p < 0.0001; PMA vs. PP2 + PMA, p < 0.0001; PMA vs. SU + PMA, p < 0.0001 (**p ≤ 0.01, ***p ≤ 0.001, ****p ≤ 0.0001). Bradykinin- and

*Figure 9 continued on next page*

*Figure 9 continued*

PMA-induced α₁1.2 tyrosine phosphorylation was blocked by PF-431396, SU6656 and PP2 but not the inactive PP3. Panel A was created using Biorender. com.

The online version of this article includes the following source data for figure 9:

**Source data 1.** Original files of the full raw unedited blots with bands labeled in red boxes.

Next we tested whether pVETL-Sh1-GFP would inhibit PMA- and BK-induced α₁1.2 phosphory-lation in PC12 cells. pVETL-Sh1-GFP lentiviral particles carrying the Sh1-shRNA and GFP expression cassettes were used to efficiently infect PC12 cells. Infected cells were monitored for GFP expression and then subjected to overnight serum starvation (to ensure low signaling levels) before treatment with PMA or BK. Fully SDS-dissociated tyrosine-phosphorylated proteins were immunoprecipitated with 4G10 before SDS–polyacrylamide gel electrophoresis (PAGE) and α₁1.2 IB (*Figure 10B, C*). As before, PMA and BK induced in average an about 2.5-fold increase in tyrosine phosphorylation of the α₁1.2 subunit, which was strongly repressed by Sh1 (*Figure 10B–D*). IB for total Pyk2 content confirmed Pyk2 knockdown by ~70–90% (*Figure 10B, C*, bottom panels). These findings indicate that depletion of Pyk2 potently blunts the PKC-mediated increase in α₁1.2 tyrosine phosphorylation. Total α₁1.2 content was not altered by Sh1. These findings indicate that Pyk2 knockdown does not affect α₁1.2 expression levels and that the reduction in tyrosine-phosphorylated α₁1.2 is likely not due to any potential off-target effects of the pVETL-Sh1-shRNA.

To further verify and extend these findings we obtained HIV-GFP lentiviral vectors for expression of validated unique 29mer shRNAs targeting Pyk2 (HIV-GFP-Pyk2ShA-D) and Src (HIV-GFP-SrcShA-D) as well as a scrambled, non-silencing control (HIV-GFP-Shscr). HIV-GFP-Pyk2ShB and C and HIV-GFP-SrcShB and C were most effective in knocking down endogenous Pyk2 and Src, respectively (*Figure 10E, F* and data not shown). PC12 cells were transduced with HIV-GFP-Pyk2ShB and C and HIV-GFP-Shscr, serum starved, stimulated with PMA, harvested, and lysed before 4G10 IP and IB for α₁1.2, Pyk2, Src, and tubulin. Total protein levels of α₁1.2, Pyk2, Src, and tubulin in lysate were moni-tored in parallel. The Pyk2-targeting HIV-GFP-Pyk2ShB and C but not the scrambled control shRNA abrogated the PMA-induced increase in α₁1.2 tyrosine phosphorylation (*Figure 10G*). Similarly, the Src-targeting HIV-GFP-SrcShB and C but not the scrambled control shRNA blocked the increase in α₁1.2 tyrosine phosphorylation upon PMA application (*Figure 10G, H*). For quantification, phosphoty-rosine signals were normalized to total α₁1.2 in lysate (*Figure 10I*). None of the HIV viral constructs exhibited any detectable effects on protein expression of α₁1.2, Pyk2, Src, or α-tubulin, vinculin, and GAPDH as determined in lysates suggesting these constructs did not affect general protein expres-sion. Collectively, the above findings indicate that knockdown effects were specific and not simply the consequence of viral infection or expression of non-specific stem-loop RNAs. Taken together, our findings strongly support the hypothesis that PKC signaling mediates its effects on Ca_V1.2 through Pyk2 and Src.

## Inhibition of Pyk2 and Src blocks LTCC-dependent LTP

Ca_V1.2 is concentrated in dendritic spines (*Hall et al., 2013*; *Hell et al., 1996*; *Leitch et al., 2009*) where it mediates Ca²⁺ influx (*Bloodgood and Sabatini, 2007*; *Hoogland and Saggau, 2004*) and several forms of LTP (*Grover and Teyler, 1990*; *Moosmang et al., 2005*; *Patriarchi et al., 2016*; *Qian et al., 2017*; *Tigaret et al., 2021*). Notably, in older mice and rats, about half of the LTP (called LTP_LTCC) induced by four 200 Hz tetani, each 0.5 s long and 5 s apart, is insensitive to NMDAR blockade but abrogated by inhibition or elimination of Ca_V1.2 (*Boric et al., 2008*; *Grover and Teyler, 1990*; *Moosmang et al., 2005*; *Shankar et al., 1998*; *Wang et al., 2016*). Pharmacological inhibition and genetic disruption of Ca_V1.2 also abolish LTP induced by either pairing presynaptic stimulation with backpropagating action potentials (*Magee and Johnston, 1997*; *Tigaret et al., 2021*; *Tigaret et al., 2016*) or by 5 Hz/3 min tetani, the latter form of LTP requiring β₂AR signaling to upregulate Ca_V1.2 activity (*Patriarchi et al., 2016*; *Qian et al., 2012*; *Qian et al., 2017*). Thus, we hypothesized that upregulation of Ca_V1.2 activity by α₁AR signaling can augment LTCC-dependent forms of LTP.

LTP_LTCC is prominent in mice older than 1 year (30–40% above baseline) but small in mice younger than 3 months (10–15% above baseline) (*Boric et al., 2008*; *Shankar et al., 1998*). LTP_LTCC requires Ca_V1.2 activity (*Moosmang et al., 2005*) and stimulation of PKC signaling via type I metabotropic

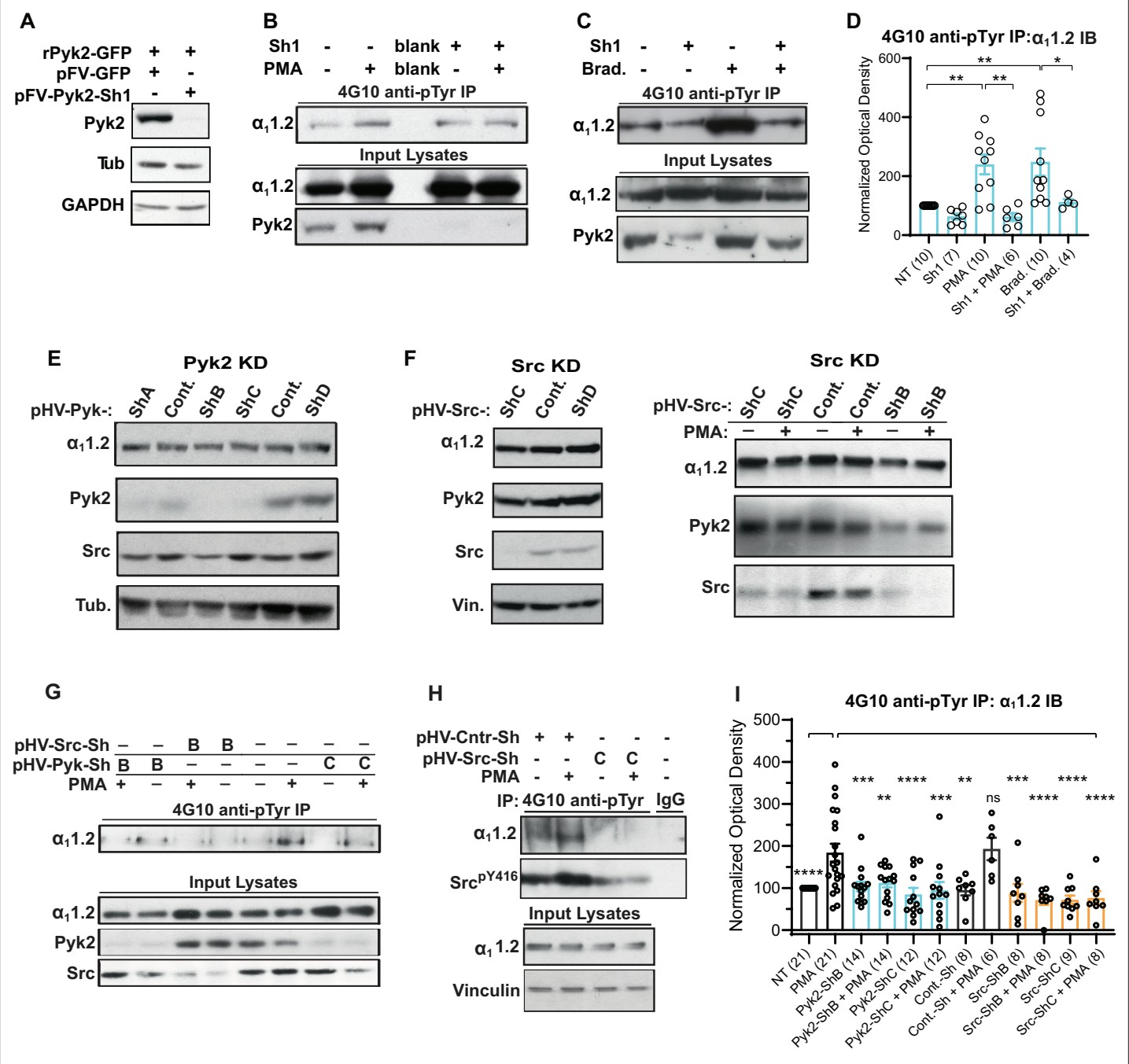

**Figure 10.** Increase in $\alpha_1 1.2$ tyrosine phosphorylation by PKC is blocked by knockdown of Pyk2 and Src. (**A**) Lysates from HEK293T/17 cells transfected with vectors encoding rat Pyk2 (rPyk2-GFP) and either the Pyk2-targeting FIV lentivirus-derived, pVETL-Sh1-GFP (pFV-Pyk2-Sh1) or control (empty) pVETL-GFP (pFV-GFP) expression vectors, were immunoblotted (IB) with indicated antibodies. (**B, C**) IB analysis of indicated proteins in PC12 cultures incubated with viral particles containing pFV-Sh1-GFP (Sh1) FIV-based expression vector used in A or medium vehicle alone for 72 hr prior to treatment with either phorbol-12-myristate-13-acetate (PMA, **B**), bradykinin (Brad.; **C**), or vehicle alone (−; **B, C**). Upper blots in B and C show anti-$\alpha_1 1.2$ IBs of 4G10-anti-phosphotyrosine (pY) immunoprecipitation (IP) while middle and lower blots show direct IBs of indicated protein levels in input lysates. (**D**) Statistical analysis of the relative pY $\alpha_1 1.2$ levels. $F_{5,41} = 8.276$. NT vs. PMA, p = 0.0031; NT vs. Brad., p = 0.0017; PMA vs. Sh1 + PMA, p = 0.001; Brad vs. Sh1 + Brad, p = 0.0433. (**E, F**) Direct IB analysis of indicated proteins in lysates of PC12 cultures transduced with HIV vector-derived lentiviral particles (e.g., pGFP-Pyk2-ShB-Lenti) containing expression cassettes for GFP and either the Pyk2-targeting (denoted pHV-Pyk-ShB and -ShC), Src-targeting (denoted pHV-Src-ShC and -ShD), or scrambled hairpin control (Cont.) shRNAs. In some cases (right blot in F) cultures were treated with PMA (+) or vehicle alone (−) before harvesting for IB. (**G, H**) IB analysis of indicated proteins from PC12 cultures infected with lentiviral particles containing HIV-GFP expression vectors as in E and F prior to treatment with either PMA (+) or vehicle (−). Upper panels show anti-$\alpha_1 1.2$ IBs of 4G10-anti-pY IP while lower

*Figure 10 continued on next page*

*Figure 10 continued*

blots show direct IBs of input lysates with indicated antibodies. (**I**) Statistical analysis of relative $\alpha_1 1.2$ pY levels. $F_{11,129}$ = 6.180. NT vs. PMA, p < 0.0001; PMA vs. Pyk2-ShB, p = 0.0005; PMA vs. Pyk2-ShB + PMA, p = 0.0029; PMA vs. Pyk2-ShC, p < 0.0001; PMA vs. Pyk2-ShC + PMA, p = 0.0002; PMA vs. Cont.-Sh, p = 0.0019; PMA vs. Cont.-Sh + PMA, p > 0.9999; PMA vs. Src-ShB, p = 0.0007; PMA vs. Src-ShB + PMA, p < 0.0001; PMA vs. Src-ShC, p < 0.0001; PMA vs. Src-ShC + PMA, p < 0.0001. The bar graphs in (**D**) and (**I**) show ratios of quantified anti-$\alpha_1 1.2$ IB signals in 4G10 IPs relative to $\alpha_1 1.2$ IB signals in total lysates, normalized to not treated (NT) control. Comparisons are made between samples treated with PMA (or bradykinin in D) and each of the other indicated conditions. Data are presented as mean ± standard error of the mean (SEM). Number (*n*) of independent experiments for each condition are indicated inside bars. Statistical analysis by analysis of variance (ANOVA) with post hoc Bonferroni's multiple comparisons test (ns = not significant vs. PMA, *p ≤ 0.05, **p ≤ 0.01, ***p ≤ 0.001, ****p ≤ 0.0001).

The online version of this article includes the following source data for figure 10:

**Source data 1.** Original files of the full raw unedited blots with bands labeled in red boxes.

glutamate receptors (mGluR) (***Wang et al., 2016***). We tested whether increasing $Ca_V1.2$ activity through $\alpha_1$AR–PKC–Pyk2–Src signaling can augment $LTP_{LTCC}$. Similar to previous reports (***Boric et al., 2008***; ***Shankar et al., 1998***), $LTP_{LTCC}$ was ~10% and was not statistically significant above baseline in our 13- to 20-week-old mice (***Figure 11A***). However, when $Ca_V1.2$ activity was upregulated by stimulation of $\alpha_1$ARs with PHE, robust $LTP_{LTCC}$ occurred (p ≤ 0.05, ***Figure 11A***). This augmentation of $LTP_{LTCC}$ was completely blocked by the LTCC inhibitor nimodipine and the $\alpha_1$AR antagonist prazosin (both p ≤ 0.001, ***Figure 11A***). Thus, this elevated potentiation strictly depends on both the activity of LTCCs and signaling through $\alpha_1$ARs. Importantly, this $LTP_{LTCC}$ is also blocked by the Pyk2 inhibitor PF-719 and the Src inhibitor PP2 (both p ≤ 0.001, ***Figure 11B***). These data indicate that robust $LTP_{LTCC}$ in 13- to 20-week-old mice requires Pyk2 and Src activity downstream of engaging $\alpha_1$AR to boost LTCC activation to sufficient levels.

## Discussion

NE is arguably the most important neuromodulator for alertness and attention, augmenting multiple behavioral and cognitive functions (***Berman and Dudai, 2001***; ***Cahill et al., 1994***; ***Carter et al., 2010***; ***Hu et al., 2007***; ***Minzenberg et al., 2008***). The $G_q$-coupled $\alpha_1$AR has a higher affinity for NE than $\beta$ARs and has been implicated in many studies in attention and vigilance (***Bari and Robbins, 2013***; ***Berridge et al., 2012***; ***Hahn and Stolerman, 2005***; ***Hvoslef-Eide et al., 2015***; ***Liu et al., 2009***; ***Puumala et al., 1997***; ***Robbins, 2002***). Inspired by our earlier findings that $Ca_V1.2$ forms a unique signaling complex with $\beta_2$AR, $G_s$, AC, and PKA, making it a prominent effector of NE (***Davare et al., 2001***; ***Patriarchi et al., 2016***; ***Qian et al., 2017***), we tested and found that $Ca_V1.2$ is also a main target for NE signaling via the $\alpha_1$AR. Given that $Ca_V1.2$ fulfills numerous functions in many cells this is a key and critical finding (***Jacquemet et al., 2016***; ***Splawski et al., 2004***). In the following paragraphs, we discuss the four central and notable outcomes of our study.

Firstly, we found that stimulation of the $\alpha_1$AR or the BK receptor strongly increased LTCC activity in neurons (***Figures 1, 2, and 4–6***). Stimulation of two other major classes of $G_q$-coupled receptors in neurons, mGluR1/5 and muscarinic M1/3/5 receptors affected LTCC activity at the cell soma only modestly or not at all, respectively (***Figure 2—figure supplement 1***). Accordingly, $G_q$-mediated signaling augments LTCC activity upon stimulation of defined but not all $G_q$-coupled receptors. Thus, activity of LTCCs is selectively regulated by $\alpha_1$AR and BK receptor signaling. Perhaps $G_q$-coupled mGluR and muscarinic receptors are not as close to the LTCCs that were recorded in somata than the $\alpha_1$AR or BK receptor, limiting their contribution to regulating $Ca_V1.2$. Defining what restricts the receptor type that can regulate LTCCs will be an interesting avenue of future investigation.

Remarkably, inhibitors of PKC, Pyk2, and Src reduce under nearly all conditions $Ca_V1.2$ baseline activity and also tyrosine phosphorylation of $Ca_V1.2$, Pyk2, and Src even when activators for $\alpha_1$AR and PKC were present. Especially notable is the strong reduction of channel activity way below the control conditions by the Src inhibitor PP2 as well as the PKC inhibitor chelerythrine in ***Figure 2C***. This effect is consistent with PP2 strongly reducing down below control conditions tyrosine phosphorylation of Src (***Figure 8J***), Pyk2 (***Figure 8L***), and $Ca_V1.2$ (***Figure 9E***) even with the PKC activator PMA present. These findings suggest that Pyk2 and Src experience significant although clearly by far not full activation under basal conditions as reflected by their own phosphorylation status, which translates into tyrosine phosphorylation of $Ca_V1.2$ under such basal conditions.

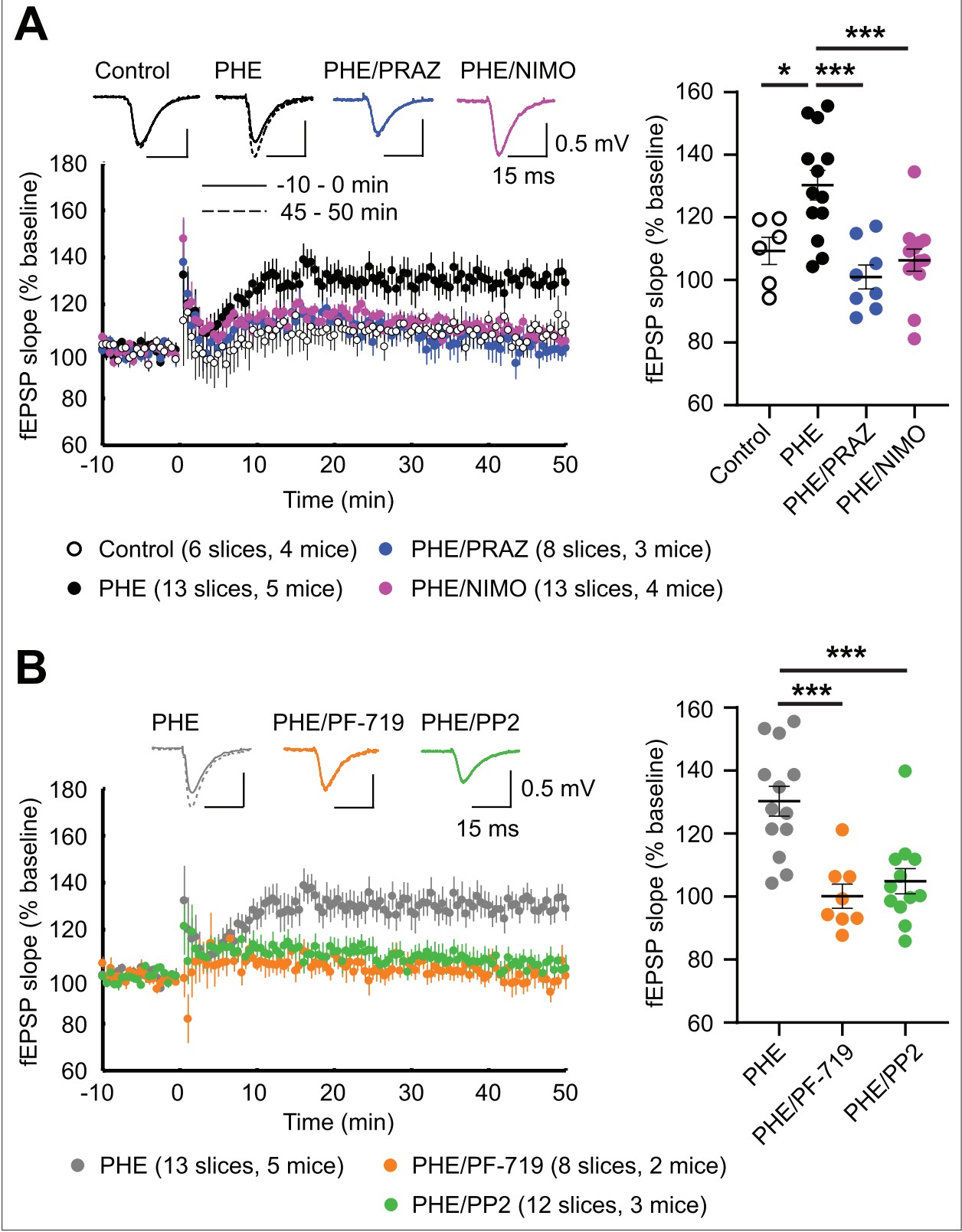

**Figure 11.** $\alpha_1$AR signaling augments $LTP_{LTCC}$ through L-type $Ca^{2+}$ channel (LTCC) activity, Pyk2, and Src. $LTP_{LTCC}$ was induced by four 200 Hz tetani, each 0.5 s long, in the CA3 Schaffer collateral projections to CA1 in acute hippocampal slices from 13- to 20-week-old mice. (**A**) $LTP_{LTCC}$ required phenylephrine (PHE; 10 μM) and was prevented by the LTCC blocker nimodipine (10 μM; NIMO) and the $\alpha_1$AR antagonist prazosin (1 μM; PRAZ). $F_{3,36}$ = 9.937. Control vs. PHE, p = 0.012; PHE vs. PHE/PRAZ, p = 0.0001; PHE vs. PHE/NIMO, p = 0.0003. (**B**) PHE-mediated long-term potentiation (LTP) is

*Figure 11 continued on next page*

Figure 11 continued

blocked by inhibitors of Pyk2 (1 μM PF-719) and Src (10 μM PP2). $F_{2,30}$ = 13.90. PHE vs. PHE/PF-719, p = 0.0002; PHE vs. PHE/PP2, p = 0.0003. Dot plots on the right show potentiation of field excitatory postsynaptic potentials (fEPSPs) determined as the averages of all responses between 45 and 50 min after high-frequency stimulation (HFS) as % of averages of all responses in the 5 min preceding HFS. Bars and whiskers represent means ± standard error of the mean (SEM; *p ≤ 0.05, ***p ≤ 0.001; one-way analysis of variance [ANOVA] with the Bonferroni correction). The number of slices and mice used is indicated.

Secondly, we identified a complex PKC/Pyk2/Src cascade that mediates regulation of Ca$_V$1.2 by the α$_1$AR. Clear evidence for this signaling pathway is provided by the inhibition of PHE-induced upregulation of LTCC activity by inhibitors of PKC, Pyk2, and Src (**Figure 2**), which is further supported by the finding that direct stimulation of PKC also upregulates LTCC activity via Pyk2 and Src (**Figure 3**). The role of the PKC/Pyk2/Src pathway in regulating Ca$_V$1.2 is also substantiated by the association of Pyk2 in addition to Src and PKC with Ca$_V$1.2 (**Figure 7**) and inhibition of PKC-induced tyrosine phosphorylation of Ca$_V$1.2 by Pyk2 and Src inhibitors (**Figure 9**) and Pyk2 and Src knockdown (**Figure 10**). Multiple shRNAs specifically targeting both Pyk2 and Src efficiently prevented the PKC-mediated increase in α$_1$1.2 tyrosine phosphorylation. These observations not only confirm that Pyk2 mediates the Ca$_V$1.2 regulation downstream of PKC but also indicates that Src itself is in this context a relevant member of the Src kinase family.

Thirdly, we found that Pyk2 is firmly associated with Ca$_V$1.2 under basal conditions as reflected by their co-IP (**Figure 7**). This association places Pyk2 into a complex that also contains its immediate upstream activator and downstream effector, that is, PKC and Src. PKC can directly bind to the distal C-terminal region of α$_1$1.2, which also contain S1928 (**Yang et al., 2005**). Given that S1928 is a phosphorylation site for PKC (**Yang et al., 2005**), it is conceivable that PKC binding to this region reflects a temporary kinase–substrate interaction rather than a more permanent association of PKC with Ca$_V$1.2, although this consideration does not rule out that PKC can stably bind to another region in the C-terminus of α$_1$1.2. In addition, the A kinase anchor protein AKAP150, which is a major interaction partner for Ca$_V$1.2 (**Davare et al., 1999**; **Hall et al., 2007**; **Oliveria et al., 2007**), binds not only PKA but also PKC (**Klauck et al., 1996**) and constitutes another potentially constitutive link between PKC and Ca$_V$1.2 (**Navedo et al., 2008**). Furthermore, like Pyk2, Src co-precipitates with Ca$_V$1.2 (**Figure 7**) and binds directly to α$_1$1.2 (**Bence-Hanulec et al., 2000**; **Chao et al., 2011**; **Hu et al., 1998**). Our determination that Pyk2 co-precipitates with Ca$_V$1.2 from not only brain but also heart indicates that Ca$_V$1.2 forms a signaling complex with Pyk2 and Src and possibly also PKC in various tissues.

We identified the loop between domains II and III of α$_1$1.2 as the binding site for Pyk2. This observation lends further support to the association of Pyk2 with Ca$_V$1.2. Of note, Src binds to residues 1955–1973 in rat brain α$_1$1.2 (corresponding to residues 1982–2000 in the original rabbit cardiac α$_1$1.2 **Mikami et al., 1989**; **Bence-Hanulec et al., 2000**; **Chao et al., 2011**). This interaction with α$_1$1.2 might bring Src in close proximity to loop II/III-associated Pyk2 to augment their structural and functional interaction once Pyk2 has been activated by PKC.

Formation of supramolecular signaling complexes or 'signalosomes' consisting of kinases and their 'customers' ensures fast, efficient, and specific signaling (**Dai et al., 2009**; **Dodge-Kafka et al., 2006**). Our work establishes the PKC–Pyk2–Src–Ca$_V$1.2 complex as such a signalosome. Furthermore, it defines how various G$_q$-coupled receptors stimulate the activity of Ca$_V$1.2 in different cells. The remarkably strong upregulation of Ca$_V$1.2 channel activity by Src (**Bence-Hanulec et al., 2000**; **Gui et al., 2006**) and upon activation of the PKC–Pyk2–Src signaling cascade as shown here rivals the upregulation by β-adrenergic signaling, which is a central and thus widely studied mechanism of regulating Ca$^{2+}$ influx into cardiomyocytes during the fight or flight response (**Balijepalli et al., 2006**; **Bean et al., 1984**; **Fu et al., 2013**; **Fu et al., 2014**; **Fuller et al., 2010**; **Lemke et al., 2008**; **Liu et al., 2020**; **Reuter, 1983**). Of note, Ca$_V$1.2 assembles all components required for β-adrenergic signaling including the β$_2$AR, G$_s$, adenylyl cyclase, and PKA in brain (**Davare et al., 2001**; **Davare et al., 1999**; **Man et al., 2020**) and heart (**Balijepalli et al., 2006**). Formation of this complex is important for upregulation of Ca$_V$1.2 activity (**Balijepalli et al., 2006**; **Patriarchi et al., 2016**) and LTP of glutamatergic synapses induced by a 5-Hz theta rhythm during β-adrenergic stimulation (**Patriarchi et al., 2016**; **Qian et al., 2017**). Analogously, assembly of the PKC–Pyk2–Src–Ca$_V$1.2 signalosome may be important for fast and specific regulation of Ca$_V$1.2 by the α$_1$AR. This hypothesis can now be tested by pursuing determination of the precise binding site of Pyk2 in the loop between domains II and III

of $\alpha_1 1.2$ and then disrupting this interaction with peptides and point mutations. However, our initial attempts to narrow down the binding region by binding studies with six synthetic ~25 residue long overlapping peptides that spanned loop II/III failed (data not shown). Perhaps the Pyk2-binding site in loop II/III requires tertiary structural elements or sequences that were distributed between two neighboring peptides (which overlapped by five residues).

In rat brain neurons, LTCC activity can be increased by Src via phosphorylation of $\alpha_1 1.2$ on Y2122 (*Bence-Hanulec et al., 2000*; *Gui et al., 2006*). However, this phosphorylation site is not conserved even within rodents. It is equivalent to position 2150 in rabbit cardiac $\alpha_1 1.2$, which is a Cys and not Tyr residue (*Mikami et al., 1989*). Accordingly, other Tyr residues must serve as phosphorylation sites. It will be an interesting challenge for future work to identify the exact phosphorylation site and then test its functional relevance.

Remarkably, the Src inhibitor PP2 also completely blocked PKC-induced autophosphorylation of Pyk2 on Y402 (*Figure 8K, L*). This finding indicates a close interdependence between Pyk2 and Src activation by PKC in PC12 cells (depicted in *Figure 8A*). It is consistent with earlier results indicating that Pyk2 activation (assessed by Y402 phosphorylation) requires catalytically active Src (*Cheng et al., 2002*; *Shi and Kehrl, 2004*; *Sorokin et al., 2001*; *Zhao et al., 2016*), although in other systems Y402 phosphorylation was not dependent on Src (*Corvol et al., 2005*; *Park et al., 2004*; *Yang et al., 2013*). Accordingly, Pyk2 autophosphorylation on Y402 and the consequent binding of Src to phosphoY402 induces Src-mediated phosphorylation of Pyk2 on Y579 or Y580 in its activation loop, which further enhances Pyk2 activity beyond the level achieved by Pyk2 autophosphorylation on Y402 (*Dikic et al., 1996*; *Lakkakorpi et al., 2003*; *Li et al., 1999*; *Park et al., 2004*). Such interdependence was supported by the observation that Y579 phosphorylation upon PKC stimulation by either PMA or BK was also completely blocked by the Src inhibitor PP2 (*Figure 8K, M*).

Fourthly and finally, $LTP_{LTCC}$ induced by 200 Hz tetani in 13- to 20-week-old mice required stimulation of $\alpha_1 ARs$ and is completely blocked by inhibitors of LTCC, Pyk2, and Src (*Figure 11*). Incidentally, we were not able to induce any $LTP_{LTCC}$ in mice younger than 13 weeks. Taken together, our findings suggest that $LTP_{LTCC}$ requires stimulation of $Ca_V 1.2$ activity by $\alpha_1 AR$–PKC–Pyk2–Src signaling. While we focus here on the importance of $\alpha_1 AR$ signaling for $LTP_{LTCC}$, this is not the only form of LTP that requires upregulation of $Ca^{2+}$ influx through $Ca_V 1.2$. Prolonged theta tetanus LTP (PTT-LTP), which is induced by a 3-min-long 5 Hz tetanus, also depends on upregulation of $Ca_V 1.2$ activity (*Boric et al., 2008*; *Cavuş and Teyler, 1996*; *Grover and Teyler, 1990*; *Moosmang et al., 2005*; *Wang et al., 2016*). In PTT-LTP, this upregulation is accomplished by $\beta_2 AR$–$G_s$–adenylyl cyclase/cAMP–PKA signaling and the ensuing phosphorylation of the central pore-forming $\alpha_1 1.2$ subunit of $Ca_V 1.2$ on S1928 by PKA (*Patriarchi et al., 2016*; *Qian et al., 2012*; *Qian et al., 2017*). Whether signaling by NE through $\alpha_1 AR$ and $\beta_2 AR$ can act in parallel and is additive will be an interesting question for future studies. However, we already know that at least for classic PTT-LTP $\beta_2 AR$ signaling is sufficient and does not require engagement of $\alpha_1 AR$ signaling (*Qian et al., 2012*). Because regulation of $Ca_V 1.2$ by $\beta_2 AR$ signaling is highly localized (*Davare et al., 2001*; *Patriarchi et al., 2016*), it is conceivable that $\alpha_1 AR$ signaling might engage a subpopulation of $Ca_V 1.2$ channels whose spatial distribution differs from that of $\beta_2 AR$-stimulated $Ca_V 1.2$ in dendrites. Alternatively, parallel engagement of $\alpha_1 AR$ and $\beta_2 AR$ signaling might ensure more robust and possibly additive or synergistic responses both at the $Ca_V 1.2$ channel level and in the synaptic potentiation that results.

LTP is thought to underlie learning and memory (*Choi et al., 2018*; *Whitlock et al., 2006*). Conditional knock out of $Ca_V 1.2$ in the hippocampus and forebrain impaired $LTP_{LTCC}$ as well as initial learning (*Moosmang et al., 2005*) and long-term memory of spatial Morris water maze tasks (*White et al., 2008*). Moreover, decreased $Ca_V 1.2$ expression or infusion of LTCC blockers into the hippocampus impaired both, LTP induced by pairing backpropagating action potentials in dendrites with synaptic stimulation and latent inhibition (LI) of contextual fear conditioning, the latter requiring learning to ignore non-relevant environmental stimuli (*Tigaret et al., 2021*). These $Ca_V 1.2$-related learning deficits might be in part due to impaired attention the animals pay to their experimental environment during learning phases, processes requiring concerted attention (*Panichello and Buschman, 2021*). Attention, in turn, depends on the neurotransmitter NE, which might augment spatial learning through regulation of $Ca_V 1.2$ via $\alpha_1 AR$–PKC–Pyk2–Src signaling.

$Ca_V 1.2$ is increasingly implicated in not just the postsynaptic physiological functions discussed above. Multiple genome-wide association studies point to variants in the $Ca_V 1.2$ gene, *CACNA1C*,

as major risk factors for schizophrenia, bipolar disorder, and other mental diseases (*Bhat et al., 2012*; *Ferreira et al., 2008*; *Green et al., 2010*; *Nyegaard et al., 2010*; *Smoller, 2013*; *Splawski et al., 2004*). Other studies link chronic upregulation of $Ca_V1.2$ activity to etiologies behind senility and Alzheimer's disease (e.g., *Davare and Hell, 2003*; *Deyo et al., 1989*; *Disterhoft et al., 1994*; *Thibault and Landfield, 1996*). Thus, it appears likely that dysfunctional regulation of $Ca_V1.2$ contributes to these diseases, making the detailed molecular analyses of the signaling paradigms that regulate its functionality especially important to advance our mechanistic understanding for development of future therapies.

Here, we establish for the first time that NE upregulates $Ca_V1.2$ activity via a complex $\alpha_1AR$ signaling cascade through PKC, Pyk2, and Src, the activity of each component being essential for $LTP_{LTCC}$ and thus most likely relevant for learning. Our work forms the foundation for future studies to uncover the physiological context in which this action of NE is specifically engaged, what the precise role for each kinase is in this signaling cascade regulating $Ca_V1.2$ activity, and how the individual kinases could be coordinately regulated to further fine-tune $Ca_V1.2$ function. Given the central role of NE in attention and the many physiological and pathological aspects of $Ca_V1.2$, regulation of this channel via NE–$\alpha_1AR$ signaling predictably will elicit widespread and profound functional effects.

# Materials and methods
## Materials availability statement
Further information and requests for resources and reagents should be directed to and will be fulfilled by the corresponding authors, Mary C. Horne (mhorne@ucdavis.edu) and Johannes W. Hell (jwhell@ucdavis.edu).

## Experimental model and subject details
### Animals
Pregnant Sprague-Dawley (SD) rats were ordered from Envigo (Order code 002) or Charles River (Strain code 001) and E18 embryos were used for preparation of dissociated hippocampal neuronal cultures. SD rats used for preparation of tissue extracts from heart and brain were of either sex and around 3 months old. For LTP experiments, mice of the strain B6129SF1/J aged between 13 and 18 weeks (both males and females) were used.

Animals were maintained with a 12/12 hr light/dark cycle and were allowed to access food and water ad libitum. All procedures followed NIH guidelines and had been approved by the Institutional Animal Care and Use Committee (IACUC) at UC Davis (Protocol # 20673 and 22403).

### Primary hippocampal neuronal cultures
Primary hippocampal neurons were maintained at 37°C in humidified incubators under 5% $CO_2$ and 95% air. Both male and female rat embryos were used to prepare the cultures. Neurons were maintained in a medium containing 1× B-27 supplement (Gibco Cat#17504044), 1× Glutamax (Gibco Cat#35050061), 5% fetal bovine serum (FBS, Corning Cat#35-010-CV), and 1 µg/ml gentamicin (Gibco Cat#15710-064) in Neurobasal medium (Gibco Cat#21103-049). 10 µM each of 5-fluoro-2'-deoxyuridine (Sigma-Aldrich Cat#F0503) and uridine (Sigma-Aldrich Cat#U3003) were added around DIV7 to block the growth of glial cells.

### Cell lines
All cells were grown at 37°C in humidified incubators under 5% $CO_2$ and 95% air. Rat pheochromocytoma cell line PC12 (ATCC Cat# CRL-1721; RRID:CVCL_0481, male) was grown in RPMI 1640 media (Gibco Cat#11875-101) containing 10% horse serum (HS, Gemini Bio Products Cat#100-508) and 5% FBS. For serum starvation, PC12 were incubated for 18 hr in RPMI 1640 media containing 1% HS and 0.5% FBS. HT-1080 cells (ATCC CCL-121; RRID:CVCL_0317, male) used for virus titration were grown and maintained in MEM (Gibco Cat#11095-080) supplemented with 10% FBS. HEK293T/17 (ATCC Cat# CRL-11268, RRID:CVCL_1926, female) cells were routinely cultured in Dulbecco's Modified Eagle Medium (DMEM) supplemented with 10% FBS (Gibco Cat#11995-065). Cell lines used were obtained from ATCC, expanded and frozen at low passage number. Care was taken during the use of cell lines to ensure that only one cell line was processed in the culture hood at a time, and that

they were used within 25–30 passages. Their morphology in culture and doubling time were routinely monitored as were other distinguishing properties such as high transfectability (HEK293T/17 cells) or high $Ca_V1.2$ expression (PC12 cells) before the time of experimental use.

### Authentication of cell lines

All three cell lines HEK293T/17, HT1080, and PC12 cells were obtained from ATCC, a well-established and highly reliable source for cell lines. These are the only cell lines currently used in our lab, minimizing further any potential for confusion.

HEK293T/17 was only used for production of virus for knockdown of Pyk2 and Src and initial testing for efficacy of respective knockdown and not for data collection. It was mostly authenticated by inspection of shape and determination of viability as well as the absence of voltage-gated ion channels including $Ca_V1.2$ as tested electrophysiologically (all hallmarks of endothelial cells like HEK293 cells). All viruses produced with this cell line were of the expected titer and infectivity as tested. Further, the knockdown results for ectopically expressed Pyk2 and Src in these HEK293T cells were consistent with subsequent knockdown of endogenous Pyk2 and Src in our PC12 cells by several viruses with respective shRNA (*Figure 9* and data not shown).

HT1080 was exclusively used for testing viral titer and mostly authenticated by inspection of shape and determination of viability.

PC12 cells are derived from a pheochromocytoma tumor and were verified in different ways. Firstly, they had the typical appearance described earlier, with some 'rugged' edges under basal conditions. Upon addition of nerve growth factor (NGF) they adopted a more neuron-like appearance with elongated protrusions reminiscent of short neurites. This response to NGF is a clear hallmark of PC12 cells and the reason why they are popular for use in biochemical experiments when neuron-like cultured cells are needed. Additional parameters were expression of L-type Ca channels as determined electrophysiologically (data not shown) and biochemically specifically for $Ca_V1.2$ as thoroughly analyzed in this study (*Figures 8–10*). In addition, expression of the BK receptor and its downstream effectors Pyk2 and Src is known to be very prominent in PC12 cells. (Pyk2 was first identified in PC12 cells) and again regularly observed in our thorough biochemical analysis.

Test for mycoplasma was performed per commercial PCR and was negative.

### Methods details

#### Culture of primary hippocampal neurons

Hippocampal neurons were cultured from wild-type E18 male and female embryos from SD rats. Hippocampus was excised from the brains of embryos in ice-cold Hank's Buffer (Sigma-Aldrich Cat#H2387) with 10 mM 4-(2-Hydroxyethyl)-1-piperazineethanesulfonic acid (HEPES) (Gibco Cat#15630-080), 0.35 g/l $NaHCO_3$ and 5 µg/ml gentamicin (Gibco Cat#15710-064) and digested in 0.78 mg/ml papain (Roche Cat#10108014001) in 5 ml of the same buffer at 37°C for 30 min, in an incubator containing 5% $CO_2$ and 95% air. Digested hippocampal tissue was washed with neuron medium twice, and triturated in the medium. The medium used for washes, trituration and culture of neurons consists of 1× B-27 supplement, 1× Glutamax, 5% FBS, and 1 µg/ml gentamicin in Neurobasal medium (as stated in Experimental model and subject details). 15,000 neurons were plated per well in 24-well plates on coverslips coated with poly-DL-ornithine (Sigma-Aldrich Cat#P0671) and laminin (Corning Cat#354232) and cultured in an incubator at 37°C and 5% $CO_2$ and 95% air.

#### Single-channel recording – determination of overall channel activity (N × Po)

Single-channel recording was performed at room temperature on hippocampal neurons on DIV10–15 using the cell-attached configuration at an Olympus IX50 inverted microscope as before (*Patriarchi et al., 2016*; *Qian et al., 2017*). The membrane potential was fixed at ~0 mV using a high $K^+$ external solution. The external (bath) solution contained (in mM) 145 KCl, 10 NaCl, 10 HEPES, and 30 D-glucose (pH 7.4 with NaOH, 325–330 mOsM). The internal (pipette) solution contained 110 mM $BaCl_2$, 20 mM tetraethylammonium chloride (TEA-Cl), 10 mM HEPES, 500 nM BayK 8644 (Tocris Cat#1546; 200 nM used for *Figures 1 and 2*; 500 nM used for *Figure 3*), and 1 µM each of $\omega$-conotoxins MVIIC and GVIA (China Peptides, custom synthesized) (pH 7.2 with TEA-OH, 325–330 mOsM). 3.5–5.5 MΩ resistance pipettes were used. The concentrations of kinase inhibitors, $\alpha_1AR$ receptor blocker and agonist are as indicated in the figure legends. Neurons were preincubated

with kinase inhibitors or receptor blocker in culture medium for 10 min prior to the experiment, and kinase inhibitors and receptor blocker were, where relevant, present in the external solution during recording. In experiments where isradipine was used, it was placed in the pipette solution, and no pre-incubation with isradipine was performed. Recordings started within 10 min of placing coverslips in the recording chamber in a bath solution with or without PHE (Sigma-Aldrich Cat#P6126), PMA (Merck Millipore Cat#524400), and different kinase inhibitors and receptor blocker. Currents were sampled at 100 kHz and low-pass filtered at 2 kHz using an Axopatch 200B amplifier (Axon Instruments) and digitized using Digidata 1440A digitizer (Axon Instruments). Step depolarizations of 2-s duration (one sweep) were elicited to the patch from −80 to 0 mV at a start-to-start stimulation interval of 7 s. Typically, 100 sweeps were recorded per neuron and only cells with more than 70 sweeps recorded were analyzed. The single-channel search event detection algorithm of Clampfit 10.7.0.3 (Axon Instruments) was used to analyze single-channel activities. Ensemble average traces were computed by averaging all sweeps from one neuron and averaging the averaged traces from all neurons in each group.

### Single-channel recording – determination of Po

To specifically determine unitary channel open probability Po, borosilicate pipettes with a resistance of 7–12 MΩ were used. Only patches with no more than 4 channels ($k \leq 4$) were included in the Po analysis to not overinterpret Po (**Horn, 1991**). Data were corrected by the number of channels ($k = 1$) as previously described (**Bartels et al., 2018**; **Turner et al., 2020**). Unitary LTCC events from hippocampal neurons (DIV15–25) where isolated through blocking N/P/Q-type calcium channels by using 1 μM each of $\omega$-conotoxins MVIIC and GVIA (China Peptides, custom synthesized) in the patch pipette and recorded as above at room temperature by step depolarizations from −80 to 0 mV. Extracellular bath solution contained 125 mM K-glutamate, 25 mM KCl, 2 mM MgCl₂, 1 mM CaCl₂, 1 mM ethylene glycol bis(2-aminoethyl)tetraacetic acid (EGTA), 10 mM HEPES, 10 mM glucose, and 1 mM Na-ATP, pH 7.4 with KOH. Depolarizing pipette solution contained 110 mM BaCl₂ and 10 mM HEPES, adjusted to a pH of 7.4 with TEA-OH. Data acquisition was performed at a sampling frequency of 10 kHz with an interpulse time of 5 s and data were low pass filtered at 2 kHz. The positive identification of LTCC activity was consequently tested by bolus application of either the dihydropyridine (DHP) BayK8644 (10 μM), which promotes L-type channel opening, or the channel-blocking DHPs isradipine (10 μM) or nimodipine (10 μM) to the bath solution at the end of each experimental run.

### Drugs

were prepared as stock solution in dH₂O, freshly on the day of experiment respectively, 40 mM prazosin–HCl (Sigma-Aldrich), 20 mM NE bitartrate (Sigma-Aldrich) and 20 mM BK acetate (Sigma-Aldrich). Stocks were diluted again 1:10 or 1:100 and as a bolus directly applied to the bath solution during the recordings.

### Co-immunoprecipitation of Ca$_V$1.2 with Pyk2 and Src

Brains and hearts were homogenized with a Potter tissue homogenizer in 10 ml of a homogenization buffer containing 50 mM Tris–HCl (pH 7.4), 150 mM NaCl, 5 mM EGTA pH 7.4, 10 mM ethylenediaminetetraacetic acid (EDTA), 1% Triton X-100, 25 mM NaF, 25 mM sodium pyrophosphate, 1 mM 4-nitrophenyl phosphate, 2 μM microcystin and protease inhibitors (1 μg/ml leupeptin [Millipore Cat#108975], 2 μg/ml aprotinin [Millipore Cat#616370], 10 μg/ml pepstatin A [Millipore Cat#516481] and 200 nM phenylmethylsulfonyl fluoride [PMSF]). High-speed centrifugation was performed at 40,000 rpm for 30 min at 4°C. 500 μg of total brain or heart lysate extracts were incubated with 15 μl of Protein-A Sepharose beads (CaptivA protein resin, Repligen, Cat#CA-PRI-0100) and 2 μg of anti-Ca$_V$1.2 α1-subunit or control rabbit IgG antibody. Samples were incubated at 4°C for 4 hr before being washed three times with ice-cold wash buffer (0.1% Triton X-100 in 150 mM NaCl, 10 mM EDTA, 10 mM EGTA, 10 mM Tris, pH 7.4). Samples were then resolved by SDS–PAGE and transferred onto polyvinylidene difluoride (PVDF) membranes before IB with anti-Ca$_V$1.2 α1-subunit (J.W. Hell lab), -Pyk2 (Millipore Cat#05-488; RRID:AB_2174219), and -Src (J.S. Brugge lab) antibodies. The antibody dilutions used are listed in **Supplementary file 2**.

## GST pulldown assay

Fragments of intracellular loops of Ca$_V$1.2 $\alpha_1$-subunit (*Supplementary file 1*; *Davare et al., 2000*; *Gao et al., 2001*) were expressed in *Escherichia coli* strain BL21 as GST fusion proteins, purified, and integrity verified by IB essentially as previously described (*Bennin et al., 2002*; *Frangioni and Neel, 1993*; *Hall et al., 2013*; *Hall et al., 2006*). Overnight cultures from single colonies of the corresponding plasmids were cultured initially in 50 ml of LB medium containing ampicillin (100 µg/ml) with aeration until saturation. Incubation temperature was optimized for each expression construct to optimize translation and stability and varied between 28 and 37°C. After a 1:10 dilution into the same medium, cultures were grown for about 2–4 hr until an A600 of about 0.8 was reached when isopropyl-β-D-thiogalactopyranoside was added for induction. After 4–5 hr bacteria were collected by centrifugation (5000 rpm, SLA 3000 rotor, Thermo Fisher Cat#07149) for 15 min and resuspended by gentle trituration in ice-cold 50 ml of Tris-buffered saline (TBS) Buffer (150 mM NaCl, 15 mM Tris-Cl, pH 7.4) containing protease inhibitors 1 µg/ml pepstatin A, 1 µg/ml leupeptin, 1 µg/ml aprotinin, and 200 nM PMSF. 0.1 mg/ml lysozyme was added to lyse cell walls. The mixture was kept on ice for 30 min before addition of Sarcosyl (1.5% final concentration), β-mercaptoethanol (10 mM), and DNAse (50 U) for 15 min to fully solubilize the fusion proteins. In order to neutralize Sarcosyl, Triton X-100 was then added to a final concentration of 5%. Insoluble material was removed by ultracentrifugation (1 hr, 4°C, 40,000 rpm, Ti70 rotor, Beckman Coulter Cat#337922). The fragments were immobilized onto glutathione Sepharose (Millipore/Cytiva Cat#17-5132-02) for 3 hr, washed three times with Buffer A (0.1% TX-100, 10 mM Tris–HCl, pH. 7.4) and incubated with affinity-purified His-tagged Pyk2 separately expressed in *E. coli* (3 hr, 4°C). Beads were washed three times in Buffer A and bound proteins were eluted and denatured in SDS sample buffer, resolved by SDS–PAGE and transferred to a nitrocellulose membrane. IB with anti-Pyk2 antibody (Millipore Cat#05-488; RRID:AB_2174219) was used to detect Pyk2 binding during pulldown.

## Analysis of phosphorylation in PC12 cells

Drugs were used at the following concentrations: 2 µM PMA (Merck Millipore Cat#524400), 1 µM BK (Sigma-Aldrich Cat#05-23-0500), 3 µM PF-431396 (Tocris Cat#4278), 10 µM PP2 (Sigma-Aldrich Cat#P0042), 10 µM PP3 (Tocris Cat#2794), and 10 µM SU6656 (Sigma-Aldrich Cat#S9692).

For phospho-tyrosine analysis PC12 cells were washed after drug treatment twice in ice-cold phosphate-buffered saline (PBS) containing 1 mM pervanadate and 25 mM NaF. Samples were collected in PBS containing pervanadate, NaF, and protease inhibitors (see above), sonicated and extracted with SDS dissociation buffer (50 mM Tris–HCl, 1% SDS) at 65°C for 10 min. The SDS was neutralized with a fivefold excess of Buffer A containing phosphatase and protease inhibitors. 500 µg of total protein from PC12 cell extracts were incubated over night at 4°C with 2 µg of the phospho-tyrosine 4G10 (Sigma-Alrich/Upstate Cat# 05-321; RRID:AB_2891016) or mouse control antibody (Jackson Immunoresearch Cat#015-000-003) and 15 µl of Protein-G Sepharose (Millipore/Cytiva Cat#GE-17-0618-05), washed three times in ice-cold wash buffer (0.1% Triton X-100 in 150 mM NaCl, 10 mM EDTA, 10 mM EGTA, 10 mM Tris, pH 7.4), resolved by SDS–PAGE and transferred onto PVDF membranes for IB. The antibody dilutions used are listed in *Supplementary file 2*. For re-probing, blots were stripped in 62.5 mM Tris-Cl, 20 mM dithiothreitol (DTT), and 2% SDS at 50°C for 30 min. Chemiluminescence immunosignals were quantified using ImageJ (*Rueden et al., 2017*) by multiple film exposures of increasing length to ensure signals were in the linear range (*Davare and Hell, 2003*; *Hall et al., 2006*). Variations in total amounts of $\alpha_1$1.2, Pyk2, and Src in the different PC12 cell lysates were monitored by direct IB of lysate aliquots. Lysate signals were used to correct $\alpha_1$1.2 signals after 4G10 IP for such variations by dividing the latter by the former.

## Lentiviral constructs for shRNA to Pyk2 and Src

A list of all shRNA target sequences is provided in *Supplementary file 3*. The shRNA sequence Sh1 against Pyk2 has been validated for Pyk2 knockdown (*Sayas et al., 2006*). The Sh1 sequence was cloned in the reverse orientation into the MfeI site of the lentiviral transfer vector pVETL-eGFP (*Bartos et al., 2010*; *Boudreau and Davidson, 2012*; *Harper et al., 2006*) for expression of Pyk2 shRNA and GFP to visualize infection. All HIV plasmids (HIV-GFP-Pyk2shA-D; HIV-GFP-SrcshA-D) for knocking down rat Pyk2 or Src as well as the scrambled, non-silencing hairpin control (HIV-GFP-shscr) were

obtained from Origene (Cat# TL710108 and #TL711639). All expression plasmids (listed in *Supplementary file 4*) were confirmed by DNA sequencing.

### Production of lentivirus for Pyk2 and Src knockdown

HEK293T/17 (ATCC Cat# CRL-11268, RRID:CVCL_1926) cells were plated onto 10 cm dishes at 1.8 × 10⁶ cells per dish and maintained until confluency (60–90%). Cells were transiently transfected with viral expression constructs using the calcium phosphate precipitation method (*Jordan et al., 1996*). For FIV virus production, cells were transfected in a 3:2:1 ratio of parental vector (pVETL, FIV 3.2): pCPRD-Env: pCI-VSVG for a total of 24 µg of DNA per plate. For production of HIV viral particles targeting Src and Pyk2, cells were transfected at a ratio of 5:2:2:2 of parental vector (e.g., pGFP-Pyk2-shC-Lenti:pCI-VSVG:pMDL g/p RRE:pRSV-REV) according to the manufacturer's guidelines (OriGene). Media was exchanged 16 hr after transfection. Media containing the packaged recombinant virus was collected at 48 and 72 hr, filtered through 0.45 µm filters, and concentrated by centrifugation (7400 × *g* for 16 hr at 4°C). The viral pellet was resuspended in ice-cold PBS, aliquoted and stored at −80 °C. Before use for transduction of PC12 cells all viral particle solutions were titered in the HT-1080 cell line (ATCC Cat#CCL-121; RRID:CVCL 0317) by seeding 12-well plates at 5 × 10⁴ cells per well 1 day before infection. 1, 5, and 10 µl of solutions containing concentrated HIV or FIV particles was added to each well and expression of GFP was monitored for 72 hr post-infection before titer was calculated.

### Slice preparation and electrophysiology

After decapitation, brains were removed from 13- to 18-week-old mice. 400-µm-thick transverse slices were made using a vibratome in cold, oxygenated (95% $O_2$ and 5% $CO_2$) dissection buffer (in mM: 87 NaCl, 2.5 KCl, 1.25 $NaH_2PO_4$, 26.2 $NaHCO_3$, 25 glucose, 0.5 $CaCl_2$, 7 $MgCl_2$, 50 sucrose). Slices were allowed to recover at room temperature for at least 1 hr in oxygenated artificial cerebrospinal fluid (ACSF) (in mM: 119 NaCl, 3 KCl, 2.5 $CaCl_2$, 1.25 $NaH_2PO_4$, 1.3 $MgSO_4$, 26 $NaHCO_3$, 11 glucose). Following recovery, slices were transferred to a recording chamber and maintained at 32–33°C in oxygenated ACSF. Field excitatory postsynaptic potential (fEPSP) was evoked by stimulating the Schaffer collateral pathway using bipolar electrode, and synaptic responses were recorded with ACSF-filled microelectrodes (1–10 MΩ) placed in the stratum radiatum of CA1 region. Recordings were acquired using an Axoclamp-2B amplifier (Axon Instruments) and a Digidata 1332 A digitizer (Axon Instruments). Baseline responses were collected at 0.07 Hz with a stimulation intensity that yielded 40–50% of maximal response. LTP was induced by four episodes of 200 Hz stimulation (0.5 s) with 5-s intervals. To measure LTCC-mediated LTP, 50 µM D-APV (Tocris Cat#0106) was included in ACSF. When used the inhibitors [10 µM nimodipine (Bayer Charge: BXR4H3P), 1 µM prazosin, 1 µM PF-719 (*Tse et al., 2012*), and 10 µM PP2] were added to ACSF from the start of the recording. PHE (10 µM) was added after at least 15 min of stable baseline, and LTP was induced ~10 min after the addition of PHE.

## Quantification and statistical analysis

Statistical analyses were performed using Prism 5 or 9 (GraphPad). Data are presented as mean ± standard error of the mean. Sample sizes, p values, and statistical tests are indicated in the figure legends. For analysis of channel NPo (*Figures 1–3*, *Figure 2—figure supplement 1*, *Figure 3—figure supplement 1*), first outliers were identified using iterative Grubb's method inbuilt in Prism. Then, statistical significance was determined using one-way analysis of variance (ANOVA) with post hoc Holm–Sidak's multiple comparisons test. For analysis of specifically Po (*Figures 4–6*), data were tested either by an unpaired or paired Student's *t*-test. For significance testing with more than two groups, a one-way ANOVA with additional Bonferroni correction was applied, p < 0.05%. For analysis of protein phosphorylation, statistical significance was determined using ANOVA with post hoc Bonferroni's multiple comparisons test (*Figures 8–10*). For analysis of LTP (*Figure 11*), a one-way ANOVA was applied followed by Bonferroni correction.

## Acknowledgements

We thank Dr. Stephen Strittmatter (Yale University) for providing PF-719. Dr. K Man executed the electrophysiological recordings shown in *Figures 1–3*; Dr. P Bartels executed the electrophysiological

recordings shown in *Figures 4–6*; Drs. Mei Shi and Mingxu Zhang performed the biochemical analysis in *Figure 7*; Dr. Peter Henderson performed the biochemical analysis in *Figures 8–10*; Dr. Karam Kim performed the LTP measurements in *Figure 11*. FUNDING This work was supported by National Institutes of Health grant R01 NS123050 (JWH), National Institutes of Health grant RF1 AG055357 (JWH), National Institutes of Health grant R01 MH097887 (JWH), National Institutes of Health grant R01 HL098200 (MFN), National Institutes of Health grant R01 HL121059 (MFN), and National Institutes of Health grant T32 GM099608 (PBH).

---

## Additional information

### Funding

| Funder | Grant reference number | Author |
|---|---|---|
| National Institutes of Health | R01 MH097887 | Johannes W Hell |
| National Institutes of Health | RF1 AG055357 | Johannes W Hell |
| National Institutes of Health | R01 HL098200 | Manuel F Navedo |
| National Institutes of Health | R01 HL121059 | Manuel F Navedo |
| National Institutes of Health | T32 GM099608 | Peter B Henderson |
| National Institutes of Health | R01 NS123050 | Madeline Nieves-Cintron |

The funders had no role in study design, data collection, and interpretation, or the decision to submit the work for publication.

### Author contributions

Kwun Nok Mimi Man, Conceptualization, Data curation, Formal analysis, Validation, Investigation, Visualization, Methodology, Writing - original draft, Project administration, Writing – review and editing; Peter Bartels, Data curation, Formal analysis, Validation, Visualization, Methodology, Writing – review and editing; Peter B Henderson, Resources, Data curation, Formal analysis, Funding acquisition, Validation, Investigation, Methodology; Karam Kim, Data curation, Formal analysis, Validation, Visualization, Methodology; Mei Shi, Data curation, Formal analysis, Validation, Investigation, Methodology; Mingxu Zhang, Data curation, Formal analysis, Investigation, Methodology; Sheng-Yang Ho, Data curation, Methodology; Madeline Nieves-Cintron, Data curation, Methodology, Writing – review and editing; Manuel F Navedo, Data curation, Funding acquisition, Methodology, Writing – review and editing; Mary C Horne, Conceptualization, Resources, Data curation, Formal analysis, Supervision, Validation, Investigation, Visualization, Methodology, Writing - original draft, Project administration, Writing – review and editing; Johannes W Hell, Conceptualization, Formal analysis, Supervision, Funding acquisition, Validation, Investigation, Visualization, Methodology, Writing - original draft, Project administration, Writing – review and editing

### Author ORCIDs

Kwun Nok Mimi Man http://orcid.org/0000-0002-0132-9129
Peter Bartels http://orcid.org/0000-0001-5852-1835
Madeline Nieves-Cintron http://orcid.org/0000-0003-1935-8400
Manuel F Navedo http://orcid.org/0000-0001-6864-6594
Johannes W Hell http://orcid.org/0000-0001-7960-7531

### Ethics

All procedures followed NIH guidelines and had been approved by the Institutional Animal Care and Use Committees (IACUC) at UC Davis (Protocol #20673 and #22403).

**Decision letter and Author response**
Decision letter https://doi.org/10.7554/eLife.79648.sa1
Author response https://doi.org/10.7554/eLife.79648.sa2

## Additional files

### Supplementary files

• Supplementary file 1. Amino acid residues of fragments of intracellular loops of Ca$_V$1.2 $\alpha_1$-subunit used in GST pulldown studies.
• Supplementary file 2. Antibody dilutions used for immunoblotting.
• Supplementary file 3. shRNA sequences targeting rat Pyk2 and Src.
• Supplementary file 4. Sequencing primers for validation of knockdown constructs.
• MDAR checklist

### Data availability

Raw datasets are available on Dryad (https://doi.org/10.25338/B86G9K).

The following dataset was generated:

| Author(s) | Year | Dataset title | Dataset URL | Database and Identifier |
|---|---|---|---|---|
| Man K, Bartels P, Henderson PB, Kim K, Shi M, Zhang M, Ho S, Nieves-Cintron M, Navedo MF, Horne MC, Hell JW | 2023 | Raw data for Manuscript entitled "Alpha-1 adrenergic receptor - PKC - Pyk2 - Src signaling boosts L-type Ca2+ channel Cav1.2 activity and long-term potentiation in rodents" | https://dx.doi.org/10.25338/B86G9K | Dryad Digital Repository, 10.25338/B86G9K |

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
