## [Editor Report]

This study reports of a new signaling pathway in hippocampal neurons by which α_1_ receptors for norepinephrine regulate Cav1.2 calcium channels; activation of α_1_ receptors enhances a form of long-lasting synaptic plasticity that is dependent on L-type calcium channels. The experiments are comprehensive and well-executed, and the main conclusions are compellingly supported by the data shown. The work has significance for the field of neuroscience in general and for cellular mechanisms of neuroregulation in particular.

---

## [Decision Letter]

**Decision letter after peer review:**

Thank you for submitting your article "α1 adrenergic receptor -PKC -Pyk2 -Src signaling boosts L-type ca^2+^ channel Cav1.2 activity and long-term potentiation in rodents" for consideration by *eLife*. Your article has been reviewed by 3 peer reviewers, and the evaluation has been overseen by a Reviewing Editor and Gary Westbrook as the Senior Editor. The reviewers have opted to remain anonymous.

The reviewers consider the study to be important in identifying a novel signaling pathway involving norepinephrine with consequences on synaptic plasticity, although some key points needing further clarifications that may require some additional experiments and/or modification of the conclusions. The reviewers have discussed their reviews with one another, and the Reviewing Editor has drafted this to help you prepare a revised submission.

Essential revisions:

1) Please provide further clarification of whether and how the linker domains II and III act in mediating Pyk2 and Src activation of Cav1.2, for example, by identifying the phosphorylation site and demonstrating its necessity by introducing a mutation.

2) Please confirm that the same pathway is targeted between PC12 cells and hippocampal neurons by testing whether bradykinin elicits the same response in hippocampal neurons.

3) Please provide additional evidence for the claim that alpha1-AR agonist increases Po by ruling out an increase in N caused e.g. by a rapid channel insertion to the membrane surface.

4) Clarify the basis for an apparent baseline modulation of channel activity by the alpha1-AR pathway that is revealed by the effects of inhibitors (Figures 2B, 2D).

In addition, please fully address all the comments raised by the three reviewers involving quantification of data, replacing of immunoblot in Figure 4B, addition of a schematic figure, and clarification/editing of the text.

*Reviewer #1 (Recommendations for the authors):*

1) My main critique would be that the study, while very well executed and rigorous, is fragmented, consisting of three parts that each feel incomplete: i, hippocampal neuron studies, mainly single channel recordings; ii, biochemical studies mainly in PC12 cells, using a different agonist bradykinin, and iii, the examination of LTP in young mice.

2) Does Norepinephrine activate this pathway in hippocampal neurons? This should be testable given the high affinity for NE of the α1-AR (line 487).

3) The single channel recordings cannot distinguish between L-type channels cav1.2 and cav1.3. This would require the use of selective knockout mice.

4) Patches identified to contain only one channel should be used to determine conclusively whether N and/or Po is increased by the α1-AR agonist Phe. This could be done by BayK8644 application at the end of the experiment, for example, as well as data mentioned at line 116. At the moment this cannot be conclusively stated that Po is increased.

5) Line 147, it is unclear why other Gq coupled receptors do not affect this pathway in hippocampal neurons. This is problematic as the study goes on to use bradykinin, rather than an α1-AR agonist in PC12 cells to dissect the pathway (Figure 5 onwards). Does bradykinin stimulate this pathway in hippocampal neurons?

6) If the same pathway is present in PC12 cells then the α1-AR should be expressed in these cells, rather than using bradykinin.

7) It is unclear why PP2 inhibits NPo and ensemble average current far below the baseline control (Figures2B and 2D), whereas SU6656 does not. This suggests non-specificity.

8) The II-III linker is identified as a binding site for Pyk2; but where are the src phosphorylation sites on CaV1.2 specifically mediating this effect? This needs to be identified, otherwise it is possible that the effect is not direct.

9) A diagram of the pathway is important to add, preferably in each figure, to show where every drug is supposed to act.

*Reviewer #2 (Recommendations for the authors):*

Overall, the work is carefully carried out but there are a few pitfalls, where information lacking. I suggest the authors to address these issues.

1. The pathway from PKC/ca^2+^, Pyk2, to Src has been already reported more than 20 years ago (see, for example, Sabri et al., Circ. Res 1998 and reference therein). But in this paper, the mechanism how noradrenaline activates Pyk2 through PKC is not known. Also the authors stated that phosphorylation of alpha11.2 by PKC inhibits Cav1.2 activity (52). Why this pathway is not active here?

2. The mechanism how Src activates Cav1.2 is not clear. The authors should make more effort to identify the site and use a mutant to show that it abolishes the increase in the channel activity. They discussed possible tyrosine phosphorylation at Y2122 but they did not confirm this. Also, this residue is located at the very end of protein, far away from the channel. It is not clear how it modulates the channel activity, if it has any function.

*Reviewer #3 (Recommendations for the authors):*

The authors should consider adding experiments that show whether the linker between domains II and III is indeed the site of regulation.

Editorial Points

Lines 203-207. These sentences are a bit garbled. Please revise.

Lines 272-273. Better wording would be.."binds via phosphoY402 to the SH2 domain".

Line 503. Better wording would be "regulate through".

---

## [Author Response]

Essential revisions:1) Please provide further clarification of whether and how the linker domains II and III act in mediating Pyk2 and Src activation of Cav1.2, for example, by identifying the phosphorylation site and demonstrating its necessity by introducing a mutation.

The relevance of Pyk2 binding to loop II/III lies in the general concept that frequently kinase signaling cascades depend on close proximity of the kinase with the target, often through direct binding. The exact binding sequence might not be close to the location of the phosphorylation site in the linear sequence because the 3-dimensional folding could bring binding and phosphorylation sites close together even if far apart in the linear sequence. Accordingly, loop II/III might not harbor the phosphorylation site, which could be anywhere in the primary sequence of the channel. Thus, identification of the phosphorylation site would likely take significantly longer than one year and without a reasonable level of guarantee of success.

However, along the lines of the Reviewers’ notion to try and define more precisely binding sites and mechanisms for further mechanistic work on Pyk2 interacting with Cav1.2 and specifically loop II/III, we ordered six overlapping fluorescein-tagged peptides that covered the whole loop II/III segment. We attempted binding studies based on fluorescence polarization, as has worked for a number of other binding sites for us in the past. Unfortunately, none of the peptides showed specific binding. Perhaps the binding region is more complex than a simple ~ 20 residue long linear segment, possibly being significantly longer or consisting of multiple short, discontinuous attachment sites that are more than several residues apart but brought together by the three-dimensional folding of loop II/III.

2) Please confirm that the same pathway is targeted between PC12 cells and hippocampal neurons by testing whether bradykinin elicits the same response in hippocampal neurons.

We used the PC12 cell system for our biochemical analysis because such biochemical analysis in hippocampal cultures is impossible due to material limitation. We simply would not be able to grow enough primary hippocampal cultures for this biochemical analysis. PC12 cells have a very strong presence of both, Cav1.2 and Pyk2. They, thus, constitute a perfect system for the biochemical analysis of Pyk2 signaling to Cav1.2. In fact, Pyk2 was first identified and characterized in PC12 cells (e.g., Lev et al., 2005: Nature 376, 737-745; Dikic et al., 2006: Nature 383, 547-549).

The rationale behind analyzing different GPCRs is that different cell types can harbor different sets of GPCRs that are coupled to Gq-PKC signaling. Stimulating Gq – PKC ‘signaling modules’ like the Gq – PKC – Pyk2 – Src module should be transferable between cell types if they share this module even if it would be engaged by different GqPCRs, like the bradykinin receptor versus α 1 AR. In addition, we went to great length in our work to also directly stimulate PKC with the phorbol ester PMA in hippocampal neurons to observe and characterize the increase in LTCC activity and in PC12 cells to observe and characterize the increase in tyrosine phosphorylation. All of these effects were inhibited by Pyk2 and Src inhibitors.

Nevertheless, expression of the Gq-coupled bradykinin receptors is quite prominent in hippocampal neurons. We tested whether their activation would also augment LTCC activity in hippocampal neurons and now report that there is a clear and strong increase in Po in Figure 6.

In these experiments we added BayK8644 at the end of the recording (as suggested by Reviewer 1, #4), which is expected to further upregulate the currents under our cell-attached patch electrode if those are mediated by LTCCs, which was consistently the case (Figure 6). This approach allowed us to confirm the identity of the channels in the patches as L-type and aided in determining the number of channels N in each patch.

3) Please provide additional evidence for the claim that alpha1-AR agonist increases Po by ruling out an increase in N caused e.g. by a rapid channel insertion to the membrane surface.

We now provide more data that support an increase in Po versus N by first forming a cell-attached seal, which isolates the small surface area from which single-channel activity is recorded, and then apply PHE. This acute application of PHE avoids delays in recording as happening when PHE is bath applied before the recording pipette is attached to the cell during which time new channels could have been inserted (Figure 4).

The acute PHE application in these new experiments induced a fast increase in the activity of Cav1.2 channels isolated under the pre-formed patch, arguing that this increase is really due to an increase in Po rather than insertion of new channels into the patch, which appears unlikely in this configuration due to spatial restrains and the time course with which the increase in channel activity happens. Furthermore, we used a recoding pipette with a much smaller diameter than in our original work (resistance of pipets used in our original recordings was 3.5-5.5 MOhm and in these new experiments it was 7-12 MOhm). The reduction in diameter results in patches that contain typically <4 channels, which is required for reliably determining the channel number N whereas the original recordings often contained >4 channels and thus cannot truly be analyzed for N versus Po. To increase our confidence in our capability to count all channels we added BayK8644 at the end of the recordings in one set of experiments (i.e., the bradykinin stimulation; Figure 6).

At the same time, and cautioned by the Reviewers’ comments, we do not want to rule out that there is also an effect of alpha1 AR signaling on channel number N in our original experiments in which cells were pre-treated with PHE or PKC activators before forming the seal. We are now using a more carefully worded interpretation of these original experiments leaving open the possibility that in these initial experiments we also could have had an effect on N. However, we would like to re-emphasize that our new data leave little room for doubt that PHE augments channel activity at least in part by increasing Po.

4) Clarify the basis for an apparent baseline modulation of channel activity by the alpha1-AR pathway that is revealed by the effects of inhibitors (Figures 2B, 2D).

We now explicitly state in the Discussion: “Inhibitors of PKC, Pyk2, and Src reduce under nearly all conditions Cav1.2 baseline activity and also tyrosine phosphorylation of Cav1.2, Pyk2, and Src even when activators for alpha1 AR and PKC were present. Especially notable is the strong reduction of channel activity way below the control conditions by the Src inhibitor PP2 as well as the PKC inhibitor chelerythrine in Figure 2C. This effect is consistent with PP2 strongly reducing down below control conditions tyrosine phosphorylation of Src (Figure 8J), Pyk2 (Figure 8L), and Cav1.2 (Figure 9E) even with the PKC activator PMA present. These findings suggest that Pyk2 and Src experience significant although clearly by far not full activation under basal conditions as reflected by their own phosphorylation status, which translates into tyrosine phosphorylation of Cav1.2 under such basal conditions.” Because there are multiple ways Pyk2 and Src can be activated including Ca influx and cell-matrix interactions, defining the cause of this baseline activity has to remain beyond the scope of the current work.

In addition, please fully address all the comments raised by the three reviewers involving quantification of data, replacing of immunoblot in Figure 4B, addition of a schematic figure, and clarification/editing of the text.

Please note that association of Src with Cav1.2 had previously been described by several authors. To ensure that the reader understands that our Src coIP with Cav1.2 is only confirmatory we now state explicitly: “We also confirmed earlier work (Figure 7A bottom panel) that indicated association of Src with Ca_V_1.2 in vitro (Bence-Hanulec et al., 2000; Endoh, 2005; Gui et al., 2006; Hu et al., 1998; Strauss et al., 1997; Wu et al., 2001) and in intact cells (Bence-Hanulec et al., 2000; Chao et al., 2011; Hu et al., 1998).” Also, we now provide the uncropped immunoblot for Figure 7B, which shows more convincingly that there is a clear band for the Src immunosignal.

Reviewer #1 (Recommendations for the authors):1) My main critique would be that the study, while very well executed and rigorous, is fragmented, consisting of three parts that each feel incomplete: i, hippocampal neuron studies, mainly single channel recordings; ii, biochemical studies mainly in PC12 cells, using a different agonist bradykinin, and iii, the examination of LTP in young mice.

We would argue that both, the single-channel data and biochemical data, are at a level of completion that is by itself appropriate for *eLife*, each characterizing a multistep signaling pathway with agonists and antagonists and also shRNA addressing each step in the signaling cascade. The LTP studies are meant to put the signaling pathway we characterized in hippocampal neurons into a larger, network level context by testing effects of key treatment conditions as established for the single-channel and tyrosine phosphorylation data.

2) Does Norepinephrine activate this pathway in hippocampal neurons? This should be testable given the high affinity for NE of the α1-AR (line 487).

We now provide data that show that application of NE after seal formation and after establishing a baseline activity also augments LTCC Po to the same extent as PHE does (compare new Figure 4A-D with new Figure 4E-H). Cav1.2 activity can also be increased by beta2 AR signaling (but not beta1 AR signaling) (Qian et al., 2017: Sci Signal 10, eaaf9659; see also Patriarchi et al., 2016: EMBO J 35, 1330-1345). This upregulation by beta2 AR stimulation is strictly mediated by localized signaling from the b2 AR to Cav1.2 and cannot be engaged when the b2 adrenergic agonist is applied from the outside of the cell attached patch formed by the recoding electrode (Davare et al., 2001: Science 293, 98-101, as cited). Thus, the increase in Po upon bath application of NE after seal formation suggests that NE as the cognate ligand for alpha1 AR can stimulate Cav1.2 activity to the same degree as PHE.

3) The single channel recordings cannot distinguish between L-type channels cav1.2 and cav1.3. This would require the use of selective knockout mice.

It is possible that the alpha1AR signaling also regulates Cav1.3 but Cav1.3 only constitutes ~20% when Cav1.2 constitutes ~80% of all L-type channels in hippocampus (Hell et al., 1993: JCB 123, 949-962; Sinnegger-Brauns et al., 2004: J Clin Invest 113, 1030-1439). Accordingly, it seems hard to imagine that the observed effects on LTCC activity could be explained solely by upregulation of Cav1.3, which would have to be extremely high to solely explain an increase in Po by 3- to 4-fold as seen in Figure 1B. With Cav1.3 only contributing 20% of the activity, this activity would have to be increased by well over 10-fold to explain an overall increase of Po of all L-type channels by twofold if the effect is solely via Cav1.3. It would be very involving to obtain conditional Cav1.2 KO mice to further prove this point and likely take well beyond one year given that we would have to import appropriate floxed mice to UC Davis (which often takes 6 months by itself, given paperwork and regulations) and then set up the breeding scheme. A full KO of Cav1.2 would in theory be easier to set up but is embryonically lethal.

4) Patches identified to contain only one channel should be used to determine conclusively whether N and/or Po is increased by the α1-AR agonist Phe. This could be done by BayK8644 application at the end of the experiment, for example, as well as data mentioned at line 116. At the moment this cannot be conclusively stated that Po is increased.

To get patches with a single channel are very rare. Figures 1, 2, and 3 are based on a total of 295 recordings under the various conditions conducted over more than 3 years. Of all of these recordings, only 21 (7%) had a single channel active at any point in time perhaps reflecting that only a single activatable channel was present in the patch. The rest (93%) of all recordings had at least 2 and typically more channels. Accordingly, it is impossible to limit analysis to patches with single channels in these experiments.

The new experiments described in response to Essential revisions point #3 (i.e., PHE increases Po when acutely applied from outside the recording electrode; Figure 4) indicates that we can detect an increase in Po when PHE is applied after seal formation. This approach complements the previous experiments when PHE was pre-applied before seal formation when more channels could be inserted into the plasma membrane during the time period between the start of the drug treatment and seal formation. In addition, we used in these new experiments recording pipettes with a smaller diameter (resistance was increased from originally 3.5-5.5 to 7-12 MOhm in these experiments), which typically yields between 1-4 channels. To confirm that the new recording conditions did typically not yield more than 4 channels, BayK8644 was added at the end of the recordings in one set of experiments (Figure 6) to ensure that our quantification of channel number N is accurate and complete. Accordingly, this approach allowed us to define patches with <4 channels for which we can reliably extract N and thereby also Po. Thus, we can now be certain that PHE specifically augments Po and not more generally NPo.

5) Line 147, it is unclear why other Gq coupled receptors do not affect this pathway in hippocampal neurons. This is problematic as the study goes on to use bradykinin, rather than an α1-AR agonist in PC12 cells to dissect the pathway (Figure 5 onwards). Does bradykinin stimulate this pathway in hippocampal neurons?

We now document that bradykinin can augment Po of LTCC in hippocampal neurons (new Figure 6). Please see response to Essential revisions point #2 for additional details.

6) If the same pathway is present in PC12 cells then the α1-AR should be expressed in these cells, rather than using bradykinin.

Systematic radioligand binding studies and functional stimulation assays did not detect any evidence for the presence of any adrenergic receptor subtypes in PC12 cells (neither alpha1, alpha2 or β AR; Williams et al., 1998: J Biol Chem 273, 24624-24632). In fact, this publication and several subsequent studies reported the use of PC12 cells to heterologously express AR subtypes to study the signaling mechanisms and functional effects of individual subtypes (e.g., Zhong and Minnemann, 1999: J Neurochem 72, 2388-2396; Olli-Lähdesmäki et al., 1999: J Neurosci. 19, 9281-9288). This strategy speaks to the relevance of the concept of ‘transposable’ signaling modules between different GqPCRs, which can engage PKC-Pyk2-Src signaling with the trimeric Gq – phospholipase C β – PKC module being the common denominator that then triggers the rest of the cascade in some but may be not all contexts.

Because our main goal is to define how Cav1.2 is regulated in neurons, for the revision we opted to test Bradykinin in neurons, as suggested by the Reviewing Editor. Also, we had tested in both experimental systems the effect of direct PKC activation by the phorbol ester PMA and of the different Pyk2 and Src inhibitors with completely congruent results (for details, please see response to point #2 by the Reviewing Editor).

7) It is unclear why PP2 inhibits NPo and ensemble average current far below the baseline control (Figures2B and 2D), whereas SU6656 does not. This suggests non-specificity.

This differential effect of PP2 versus SU6656 could hint that Src is not the only Src family kinase (SFK) that is involved with other SFKs perhaps also playing a role as they have different sensitivities to PP2 and SU6656 (see, e.g., Blake et al., 2000: Mol Cell Biol. 20, 9018-9027; Bain et al., 2007: Biochem J 408, 297-315). However, this effect was not consistently observed. In detail, the PP2 effect is not as strong and the SU6656 stronger in Figure 3 compared to Figure 2 (although both inhibitors reduce Po below baseline levels in both figures). These observations are potentially reflecting some experimental or biological variability between experiments as these data were collected over a long time period (>3 years) because these experiments are very time consuming and dependent on, among other factors, obtaining good hippocampal cultures and having very low electric noise in the recording cage, both of which are not trivial. This variability provides an alternative explanation to the notion that another SFK member could be involved.

8) The II-III linker is identified as a binding site for Pyk2; but where are the src phosphorylation sites on CaV1.2 specifically mediating this effect? This needs to be identified, otherwise it is possible that the effect is not direct.

Although Gq-PKC-Pyk2-Src signaling clearly induces tyrosine phosphorylation of the Cav1.2 alpha1 subunit, we cannot exclude that our effects on Po are indirect, i.e., through phosphorylation of another subunit or channel component. However, we feel that the Po effects are of strong functional interest, whether due to direct or indirect phosphorylation. The biochemical work provides now impetus that justifies testing whether phosphorylation of Cav1.2 is mediating the functional effect (we are currently applying for funding for continuation of this project). However, given the already huge time commitment of the current work it seems outside the scope of the current project to determine the relevant phosphorylation sites, which can take years.

As stated in response to Essential revisions point #1 we were not able to define a synthetic peptide derived from loop II/III that would bind Pyk2 and thus cannot use a peptide displacement approach to acutely test the relevance of Pyk2 binding to Loop II/III.

9) A diagram of the pathway is important to add, preferably in each figure, to show where every drug is supposed to act.

We now added a scheme to each figure to make it easier for readers to follow the pathways and experiment design and data interpretation.

Reviewer #2 (Recommendations for the authors):Overall, the work is carefully carried out but there are a few pitfalls, where information lacking. I suggest the authors to address these issues.1. The pathway from PKC/ca^2+^, Pyk2, to Src has been already reported more than 20 years ago (see, for example, Sabri et al., Circ. Res 1998 and reference therein). But in this paper, the mechanism how noradrenaline activates Pyk2 through PKC is not known. Also the authors stated that phosphorylation of alpha11.2 by PKC inhibits Cav1.2 activity (52). Why this pathway is not active here?

In addition to Sabri et al., 1998, the PKC-Pyk2-Src signaling cascade was first identified in PC12 cells (Lev et al., 1995: Nature 376, 737-745; Dikic et al., 1996: Nature 383, 547-549) but no previous work linked this signaling cascade to Ca channel regulation. The primary importance of our work is to define whether and how alpha1 AR signaling regulates Cav1.2 in neurons because norepinephrine is a key neuromodulator that governs attention and, at higher levels, stress responses. The role of alpha1 AR in regulating neuronal Cav1.2 is indicated by our finding that the alpha1 AR-selective agonist PHE increases Po and that this increase is blocked by the alpha1 AR-selective antagonist prazosin. Another important aspect is to define regulation of Cav1.2 by the PKC-Pyk2-Src cascade with PKC, Pyk2, and Src forming a signaling complex with Cav1.2 on a functional and biochemical level as discussed in the manuscript.

As we now point out explicitly in the Introduction, the sites for direct phosphorylation of Cav1.2 by PKC (S27 and S31) that inhibit channel activity (McHugh et al., 2000) (former Ref 52) are located in a differentially spliced exon that is prominently expressed in heart but not neurons as further detailed by (Snutch et al., 1991) to explain why PKC does not inhibit Cav1.2 in neurons. We state: “T27/T31 are not present in the most prevalent brain isoform due to alternative splicing (Snutch et al., 1991), thus the inhibitory effect of PKC on LTCC currents is typically absent in neurons and neural crest-derived PC12 cells, or in vascular smooth muscle (Navedo et al., 2005; Taylor et al., 2000).”

2. The mechanism how Src activates Cav1.2 is not clear. The authors should make more effort to identify the site and use a mutant to show that it abolishes the increase in the channel activity. They discussed possible tyrosine phosphorylation at Y2122 but they did not confirm this. Also, this residue is located at the very end of protein, far away from the channel. It is not clear how it modulates the channel activity, if it has any function.

Please see response to Essential revisions point #1 and point #7 of Reviewer 2. Y2122 is only seen in rodents but not present in other mammals including humans. Y2122 is, thus, unlikely to be a general major regulatory site, which we now discuss more explicitly. Furthermore, as the Reviewer points out, it is very distal to the channel. Finally, perhaps we should emphasize that determining regulation of Cav1.2 by either cAMP/PKA or Gq/PKC signaling has been hampered for the last 3 decades in the many different labs that had been working on these issues by not being able to consistently and reproducibly being able to reconstitute either regulatory mechanism for Cav1.2 in HEK293 or other cell lines (mostly personal communications from multiple PIs but see some primary data and discussion of this issue in our review Dai, Hall, and Hell 2009: Physiol Rev 89, 411-452; p420, left column; see also Man, Bartels, Horne, and Hell, 2020: Sci Signal 13, eabc6438). Accordingly, we cannot readily express WT and mutant Cav1.2 (Iike Y2122F) in HEK293 cells and test whether regulation by PKC-Pyk2-Src is affected or not.

Reviewer #3 (Recommendations for the authors):The authors should consider adding experiments that show whether the linker between domains II and III is indeed the site of regulation.

Please see response to Essential revisions point #1 and #8 by Reviewer 1. Briefly, we attempted but were not able to identify shorter loopII/III-derived peptides that would constitute the Pyk2 binding site.

Editorial PointsLines 203-207. These sentences are a bit garbled. Please revise.

We revised this section mostly by simplifying the statement, which now reads: “Kinases and proteins that regulate kinase activity are often found in complexes with their ultimate target proteins (i.e., their ultimate substrates) including different ion channels for efficient and specific signaling (Dai *et al.*, 2009; Dodge-Kafka *et al.,* 2006).”

Lines 272-273. Better wording would be …"binds via phosphoY402 to the SH2 domain".

We revised this statement accordingly.

Line 503. Better wording would be "regulate through".

We revised this statement accordingly.